# Overparametrization bends the landscape: BBP transitions at initialization in simple Neural Networks

**Brandon Livio Annesi**[*,1]    **Dario Bocchi**[*,1,2]    **Chiara Cammarota**[1,3]

[1]Physics Department, University of Rome 'La Sapienza', Piazzale Aldo Moro 5, Rome, 00185, Italy
[2]Institute of Nanotechnology, CNR-NANOTEC, Piazzale Aldo Moro, 5, 00185 Rome RM
[3]National Institute for Nuclear Physics, INFN-Roma1, Piazzale Aldo Moro 5, Rome, 00185, Italy
{brandonlivio.annesi, dario.bocchi, chiara.cammarota}@uniroma1.it

## Abstract

High-dimensional non-convex loss landscapes play a central role in the theory of Machine Learning. Gaining insight into how these landscapes interact with gradient-based optimization methods, even in relatively simple models, can shed light on this enigmatic feature of neural networks. In this work, we will focus on a prototypical simple learning problem, which generalizes the Phase Retrieval inference problem by allowing the exploration of overparametrized settings. Using techniques from field theory, we analyze the spectrum of the Hessian at initialization and identify a Baik–Ben Arous–Péché (BBP) transition in the amount of data that separates regimes where the initialization is informative or uninformative about a planted signal of a teacher-student setup. Crucially, we demonstrate how overparameterization can *bend* the loss landscape, shifting the transition point, even reaching the information-theoretic weak-recovery threshold in the large overparameterization limit, while also altering its qualitative nature. We distinguish between continuous and discontinuous BBP transitions and support our analytical predictions with simulations, examining how they compare to the finite-N behavior. In the case of discontinuous BBP transitions strong finite-N corrections allow the retrieval of information at a signal-to-noise ratio (SNR) smaller than the predicted BBP transition. In these cases we provide estimates for a new lower SNR threshold that marks the point at which initialization becomes entirely uninformative.

## 1    Introduction

The geometry of high-dimensional, non-convex loss, risk, or cost landscapes plays a central role in modern machine learning and data science. Such landscapes hide important structural features of the data into specific local structures and mostly deep configurations. Despite their complexity, the optimization of these landscapes is typically performed using local iterative algorithms, most notably gradient descent and its stochastic variants. Understanding the success and limitations of these algorithms remains a fundamental open problem. It has been first observed that in regimes where the dataset is large relative to the problem dimension $N$, *i.e.*, at high signal-to-noise ratio (SNR), the landscape can undergo an effective trivialization, becoming nearly convex and devoid of spurious local minima (Fyodorov, 2004; Soudry & Carmon, 2016; Cai et al., 2022). In this setting, each point in the landscape contains a clear directional signal guiding the optimization toward informative minima. Furthermore, it has been widely reported that overparameterization of the learning model can induce a smoothing of the loss landscape even in regimes with moderate or low SNR, thereby facilitating optimization (Shevchenko & Mondelli, 2020; Cooper, 2021). Perhaps more surprisingly, even in settings where spurious non-informative minima remain prevalent, gradient-based methods often still succeed (Baity-Jesi et al., 2018; Liu et al., 2020; Ros et al., 2019; Mannelli et al., 2019). This apparent paradox has been addressed in a series of works on high-dimensional inference problems such as matrix-tensor PCA (Sarao Mannelli et al., 2019) and phase retrieval (Sarao Mannelli

---
[*]Equal contribution

et al., 2020a). These studies reveal that gradient flow dynamics can avoid these poor solutions due to the local geometry of high-dimensional basins of attraction, which are typically explored by the dynamics. The high dimensional basins of attraction of gradient flow, although still non informative themselves, develop an instability towards the signal at relatively low SNR. This phenomenon is sometimes referred to as the "blessing of dimensionality". Crucially, the emergence of such instabilities at increasing SNR is associated with a qualitative change in the spectrum of the local Hessian in a transition known as the Baik–Ben Arous–Péché (BBP) transition (Baik et al., 2005).

Alternative learning approaches, mostly applied to signal reconstruction problems, are based on the use of spectral methods (Netrapalli et al., 2013; Montanari & Sun, 2018) to define a warm start to subsequent local iterative algorithms, with the aim of boosting their performances. Typically, in spectral methods such initial guess is provided by the leading eigenvector of a matrix, which is a function of the input data tailored to the structure of the specific problem (Montanari & Sun, 2018; Lu & Li, 2020; Mondelli & Montanari, 2018; Maillard et al., 2022; Defilippis et al., 2025; Kovačević et al., 2025). Interestingly in some cases, for instance phase retrieval or tensor PCA, such *ad hoc* procedure can be also linked to the risk landscape as the matrix used for spectral initialization corresponds to the negative Hessian of a suitably defined cost function evaluated at random configurations and averaged over many of them (Biroli et al., 2020). Therefore its leading eigenvector represents the direction with the most negative (or smallest positive) curvature found at the initial condition in such averaged landscape. Also in this case, when the SNR increases, the spectrum of such matrix undergoes a BBP transition, after which the leading eigenvector develops a finite correlation with the signal. A similar phenomenon occurs in the Hessian of the cost function evaluated at individual random configurations (Bonnaire et al., 2025; Arous et al., 2025), but it is less known how this landscape feature is also affected by overparametrization. Moreover, very recent work (Bonnaire et al., 2025) has shown that the information contained in the curvature of the landscape in random configurations, for finite input dimensions $N$, could further automatically help gradient-based methods in finding the deep informative minima. The interplay between the gradient flow algorithmic transitions and the emergence of the signal in the Hessian at the initial condition then defines an effective algorithmic transition in the SNR, which slowly changes with the dimensionality of the data set. This already nontrivial mechanism may be further modified in the presence of overparameterization, motivating a deeper exploration of its role in shaping the optimization landscape and the dynamics therein.

In this work, we consider an extended version of the classical phase retrieval problem by focusing on a teacher-student setting based on two-layer soft-committee machines with quadratic activations. The widths of the hidden layers of the student and teacher networks, denoted by $p$ and $p^*$ respectively, are generic and finite, while the dimensionality $N$ of the input samples will be considered very large and diverging, except in numerical tests. When $p = p^* = 1$, the setting reduces to the standard phase retrieval problem, which involves recovering a hidden signal from magnitude-only projections. It notoriously results in a non-convex optimization problem with broad relevance in optics (Millane, 1990), signal processing (Bendory et al., 2017), quantum mechanics (Orl et al., 1994), and which has often served as a prototypical example for exploring the interplay between optimization dynamics and high-dimensional geometry (Sun et al., 2018).

For general $p$ and $p^*$, this setting is a particular case of what is known in the literature as a *multi-index model* Li (1991); Defilippis et al. (2025); Troiani et al. (2024). Here we focus on the case $p > p^*$, and explore the effect of overparametrization on the landscape structure. In particular, we focus on the information contained in the local curvature in random positions of a suitably defined class of loss landscapes spanned by a parameter $a$. We study how it changes with $a$, $p$ and $p^*$. As previously mentioned, the Hessian at initialization could contain more information than expected, which could lower the SNR of algorithmic transitions for gradient-based algorithms, or could be explicitly used in a sort of generalized spectral method.

Our analysis shows that overparametrization generally shifts the BBP transition in the Hessian spectra of random configurations toward lower SNR. The corresponding spectral initialization method based on such local Hessian matrices is therefore expected to extract information earlier than in the underparametrized case and even gradient-based learning dynamics is expected to work better with overparametrization in finite-dimensional practical implementations of the problem. However, we also obtain that in few very specific instances overparametrization may slightly harm the efficiency of signal recovery obtained through the diagonalization of the Hessian at initialization. We also observe that the nature of the BBP transition changes from underparametrized students to students

benefitting from overparametrization. When overparametrization increases the standard BBP transition tends to be replaced by a BBP transition associated with a discontinuous jump in the amount of information retrieved. The emergence of discontinuous BBP transitions has been only very recently discussed in association to signal reconstruction in phase retrieval problems (Bocchi et al., 2026a; Bousseyroux & Potters, 2024), and previously only conjectured on theoretical grounds (Potters & Bouchaud, 2020). With this work, we illustrate how they become central when phase retrieval is generalized to an overparametrized learning setup. Moreover, strong finite size effects are expected to affect numerical observation of discontinuous BBP transitions (Bocchi et al., 2026a). We highlight this aspect in the results of the signal recovery in the overparametrized cases. Interestingly, we observe that higher overparametrization renders the transition more discontinuous. This effect tends to counterbalance the small shift to higher SNR of the signal recovery transition—in the large dimensional limit—obtained at higher overparametrization in specific instances, effectively reinstating a generalized advantage of overparametrization in realistic applications. Finally, we discuss the large overparametrization limit $p \to \infty$ for fixed $p^*$ and we reobtain the weak recovery algorithmic transition at an SNR equal to $p^*/2$, already discussed in the literature for $p^* = 1$ in Mondelli & Montanari (2018) and for $p^* > 1$ in the Bayes optimal setting in Maillard et al. (2024); Troiani et al. (2024).

## 2 THE MODEL

The teacher–student framework provides a simplified yet powerful setting for studying supervised learning. In this setup, a student network $\hat{y}(\mathbf{x}) : \mathbb{R}^N \to \mathbb{R}$ is trained to match the outputs of an unknown teacher function $y(\mathbf{x}) : \mathbb{R}^N \to \mathbb{R}$ using a set of $M = \alpha N$ labeled examples. In this model the parameter $\alpha$, which controls the size of the dataset compared to the input size, acts as the SNR described in the previous section. The examples consist of input vectors $\{\mathbf{x}^\mu\}_{\mu=1}^M$ drawn independently from a Gaussian distribution $\mathcal{N}(0, \mathbb{I}_N/N)$, along with their corresponding teacher outputs $y^\mu = y(\mathbf{x}^\mu)$. In the case of soft-committee machines with quadratic activations, for a given input vector $\mathbf{x}^\mu$, the output of the teacher network is:

$$y(\mathbf{x}^\mu) = \frac{1}{p^*} \sum_{l=1}^{p^*} (\mathbf{w}_l^* \cdot \mathbf{x}^\mu)^2 \equiv \frac{1}{p^*} \sum_{l=1}^{p^*} (u_l^\mu)^2 , \tag{1}$$

where $p^*$ is the width of the hidden layer of the teacher network and $u_l^\mu \equiv \mathbf{w}_l^* \cdot \mathbf{x}^\mu$ is the pre-activation output of the $l$-th teacher node $\mathbf{w}_l^*$. Each teacher node $\mathbf{w}_l^*$ is independently sampled from the sphere $S_{N-1}(\sqrt{N})$. Similarly, the output of the student network is defined as:

$$\hat{y}(\mathbf{x}^\mu) = \frac{1}{p} \sum_{k=1}^{p} (\mathbf{w}_k \cdot \mathbf{x}^\mu)^2 \equiv \frac{1}{p} \sum_{k=1}^{p} (\lambda_k^\mu)^2 , \tag{2}$$

where $p$ is the width of the hidden layer of the student network and $\lambda_k^\mu \equiv \mathbf{w}_k \cdot \mathbf{x}^\mu$ is the pre-activation output of the $k$-th student node $\mathbf{w}_k$. Standard gradient descent algorithms iteratively modify the weights $\{\mathbf{w}_k\}_{k=1}^p$ to minimize an empirical loss on the training data $\{\mathbf{x}^\mu\}_{\mu=1}^M$. Following previous works (Bonnaire et al., 2025), we define a family of normalized quadratic loss functions:

$$\mathcal{L}_\mathbf{w} = \sum_{\mu=1}^{M=\alpha N} \ell_\mathbf{w}(\mathbf{x}^\mu) \equiv \frac{1}{2} \sum_{\mu=1}^{M=\alpha N} \frac{[y(\mathbf{x}^\mu) - \hat{y}(\mathbf{x}^\mu)]^2}{a + y(\mathbf{x}^\mu)} , \tag{3}$$

where the parameter $a > 0$ controls the strength of the normalization that prevents pathologies due to rare very small or very large teacher outputs. By regulating the conditioning of the Hessian eigenspectrum, the denominator ensures the appearance of a finite left edge, an essential feature for our analytical analysis focusing on an isolated eigenvalue exiting from the left.

Instead of studying the dynamics of the learning process, for which we only provide preliminary results in Appendix F, we focus here on the structure of the loss landscape itself. In particular, we are interested in the local curvature of the empirical loss at initialization, which is governed by the spectral properties of its Hessian matrix $\mathcal{H} \in \mathbb{R}^{pN \times pN}$. This can be seen as a block matrix, comprising of $p^2$ blocks of $N \times N$ matrices $\mathcal{H}_{qq'}$, defined as

$$(\mathcal{H}_{qq'})_{ij} = \frac{\partial^2}{\partial(\mathbf{w}_q)_i \, \partial(\mathbf{w}_{q'})_j} \sum_{\mu=1}^{\alpha N} \ell(\{u_l^\mu\}, \{\lambda_k^\mu\}, \{\mathbf{x}^\mu\}) \equiv \sum_{\mu=1}^{\alpha N} F_{qq'}^\mu x_i^\mu x_j^\mu , \tag{4}$$

$$\text{where} \quad F_{qq'}^{\mu} = \frac{2}{p} \cdot \frac{\frac{2}{p}\lambda_q^{\mu}\lambda_{q'}^{\mu} + \delta_{qq'}\left[\frac{1}{p}\sum_{k=1}^{p}\left(\lambda_k^{\mu}\right)^2 - \frac{1}{p^*}\sum_{l=1}^{p^*}\left(u_l^{\mu}\right)^2\right]}{a + \frac{1}{p^*}\sum_{l=1}^{p^*}\left(u_l^{\mu}\right)^2}, \tag{5}$$

where the pre-activations $\{\lambda_k^{\mu}\}$ and $\{u_l^{\mu}\}$ are random iid variables $\mathcal{N}(0,1)$.

Let $\{h_i\}$ denote the eigenvalues of $\mathcal{H}$. In the large-$N$ limit, the spectrum consists of a continuous "bulk" component, described by the density

$$\rho(\lambda) = \lim_{N \to \infty} \frac{1}{pN} \sum_{i=1}^{pN} \delta(\lambda - h_i), \tag{6}$$

along with a finite number of outlier eigenvalues. From these, one can extract information about the geometry of the loss landscape, such as the presence of directions correlated with the signal. This procedure can be connected to a broader class of techniques known as *spectral methods* (Lu & Li, 2020; Mondelli & Montanari, 2018; Maillard et al., 2022). These approaches are based on constructing matrices of the form

$$\mathcal{D} = \sum_{i=1}^{\alpha N} T\big(y(\mathbf{x}^{\mu})\big)\mathbf{x}^{\mu}(\mathbf{x}^{\mu})^T, \tag{7}$$

where $T : \mathbb{R} \to \mathbb{R}$ is an appropriate pre-processing function, to study the simple phase retrieval problem (which in our notation corresponds to the $p = p^* = 1$ case). The leading eigenvector, *i.e.* the eigenvector associated with the largest or smallest eigenvalue, depending on the sign convention, is then computed to provide an estimate of the underlying signal. This estimate can be used directly as a proxy for the signal or serve as an initialization for a subsequent descent-like optimization algorithm.

Whether this spectral reconstruction successfully aligns with the true signal depends on the signal-to-noise ratio $\alpha$. This phenomenon is captured by the Baik–Ben Arous–Péché (BBP) transition (Baik et al., 2005), which describes a phase boundary in the spectrum: only when the signal-to-noise ratio exceeds a critical threshold $\alpha_{BBP}$ does a leading eigenvalue detach from the bulk of the spectrum, allowing its associated eigenvector to carry non-trivial information about the signal. Below this threshold, the spectrum remains uninformative, and the leading eigenvector fails to align with the teacher.

Interestingly, for $p = p^* = 1$ the forms of matrices $\mathcal{H}$ and $\mathcal{D}$ are similar, the main difference being that the pre-processing function $T$ only depends on the labels $y(\mathbf{x}^{\mu})$ while the factors $F_{11}^{\mu}$ depend both on the labels and the student outputs $\hat{y}(\mathbf{x}^{\mu})$. For the right function $T$ however, $\mathcal{H}$ can be mapped into $\mathcal{D}$ by averaging over the student weights $\mathbf{w}$. Spectral methods then can be interpreted as extracting information from this averaged Hessian. This was first noted in Biroli et al. (2020), where authors use this perspective to develop a spectral method for a different inference problem called *tensor PCA*.

In this work, rather than studying the averaged Hessian, we study the spectral properties of the actual Hessian, following the lines of Bonnaire et al. (2025). We extend this analysis to the more general case of arbitrary student and teacher widths $(p, p^*)$. Specifically, in this work we study the BBP transition of the training loss Hessian at initialization, i.e., when the student network weights are randomly and independently sampled from the sphere $S_{N-1}(\sqrt{N})$, for a teacher with a generic number of nodes ($p^* \geq 1$), and examine the effect of student overparameterization ($p > p^*$). In some sense, overparameterization can be viewed as implicitly averaging the loss landscape across the many student nodes. Indeed, we will show that in the limit of infinite overparameterization, the performance converges to that of the optimal spectral method found in Mondelli & Montanari (2018) for $p^* = 1$.

To build intuition for how the BBP transition extends beyond the phase retrieval setting, we begin by recalling the simpler case. In phase retrieval, for signal-to-noise ratios larger than a critical threshold $\alpha_{BBP}$ a single eigenvalue $\lambda^*$ separates from the bulk of the spectrum, and its associated eigenvector $\mathbf{v}^*$ exhibits nontrivial alignment with the signal vector $\mathbf{v}$. This alignment is quantified by the normalized overlap

$$m = \frac{\mathbf{v}^* \cdot \mathbf{v}}{\|\mathbf{v}^*\|\|\mathbf{v}\|}. \tag{8}$$

In the more general two-layer teacher setting, isolated eigenvalues similarly correspond to alignments between student and teacher nodes. The student output can be rewritten as

$$\hat{y}(\mathbf{x}^\mu) = (\mathbf{x}^\mu)^\top \frac{\mathbf{W}^\top \mathbf{W}}{p} \mathbf{x}^\mu; \quad \mathbf{W} \in \mathbb{R}^{p \times N}, \; W_{ki} = (\mathbf{w}_k)_i, \tag{9}$$

where the matrix $\mathbf{W}$ collects the student weight vectors $\mathbf{w}_k$. This expression is invariant under rotations $\mathbf{W} \mapsto \mathbf{O}\mathbf{W}$ with $\mathbf{O} \in \mathbb{R}^{p \times p}$ an orthogonal matrix, implying that the learned configuration is only identifiable up to orthogonal transformations (Sarao Mannelli et al., 2020b; Martin et al., 2024; Bocchi et al., 2026b). For an eigenvector $\mathbf{v}^{kl}$ associated with alignment between student node $k$ and teacher node $l$, the overlap is defined as:

$$m_{kl} = \sqrt{\sum_{i=1}^{p} \sum_{j=1}^{p^*} \left(M_{ij}^{kl}\right)^2}, \tag{10}$$

where $M^{kl} = \mathbf{V}^{kl}(\mathbf{W}^*)^\top$, $\mathbf{W}^*$ collects the teacher weight vectors $\mathbf{w}_k^*$ and $\mathbf{V}^{kl} \in \mathbb{R}^{p \times N}$ is the reshaped eigenvector matrix. This represents the Frobenius norm of the overlap matrix between the eigenvector and teacher weights. Since all student–teacher overlaps are equivalent, they can be summarized by a single scalar parameter $m = m_{kl} \; \forall k, l$. Further details can be found in appendix C.

## 3 METHODOLOGICAL OVERVIEW

In this section we give a brief overview of the analytical methods used to compute the spectrum of the matrix $\mathcal{H}$ in the case $p = p^* = 1$. We refer to the appendix the full details of the computation, and how it can be extended to calculate $\alpha_{BBP}$. Our analysis is based on field-theoretic techniques first introduced in Zee (1996), but rarely employed in the Statistical Physics/Machine Learning community. The objective is the computation of the Stieltjes transform of the spectral distribution

$$g(z) = \lim_{N \to \infty} \mathbb{E}_{\boldsymbol{x}} \frac{1}{N} \text{Tr}\left(\frac{1}{z\mathbf{I} - \mathcal{H}}\right). \tag{11}$$

This quantity can be cast in the field-theoretic formalism by introducing an $N$-dimensional scalar field $\boldsymbol{\psi}$ and using a basic identity for Gaussian integration to write

$$g(z) = \lim_{N \to \infty} \mathbb{E}_{\boldsymbol{x}} \frac{1}{\mathcal{Z}} \int d\boldsymbol{\psi} \, e^{-\frac{1}{2}\boldsymbol{\psi}^T(z\mathbf{I} - \mathcal{H})\boldsymbol{\psi}} \frac{\|\boldsymbol{\psi}\|^2}{N} \tag{12}$$

where $\mathcal{Z}$ is the normalization constant. This integral cannot be computed exactly, however we can expand the exponential $e^{-\frac{1}{2}\boldsymbol{\psi}^T \mathcal{H} \boldsymbol{\psi}}$, and take the average of every term with respect to the gaussian measure of the fields $\boldsymbol{\psi}$ and $\{\boldsymbol{x}^\mu\}_{\mu=1}^{P}$. According to Wick's probability theorem, these averages can be expressed as the sum over all possible pairings between fields of their covariances. For example the average

$$\langle \psi_i \psi_k x_k^\mu x_l^\mu \psi_l \psi_j \rangle = \langle x_k^\mu x_l^\mu \rangle \left( \langle \psi_i \psi_k \rangle \langle \psi_l \psi_j \rangle + \langle \psi_i \psi_l \rangle \langle \psi_k \psi_j \rangle + \langle \psi_i \psi_j \rangle \langle \psi_k \psi_l \rangle \right) \tag{13}$$

To track such combinations, a graphical method due to Feynman is used, and each term is expressed as a diagram. We represent each $\langle \psi_i \psi_j \rangle$ with a straight black line, and every $\langle x_i^\mu x_j^\mu \rangle$ with a double blue line. For example, the first term in the expansion can be written in a diagram form as

Weight: $\frac{1}{N^2} \frac{1}{z^2} \sum_{\mu,i} F_{11}^\mu$,

By understanding the general form of these diagrams, it is possible to rule out a whole family of subdominating ones, and to express the Stieltjes transform as a function of the sum of a certain type of diagrams, that are called in Physics *1-Particle Irreducible* (1PI) diagrams. If we call $\Sigma(z)$ the sum of all such diagrams, then we get the final expression

$$g(z) = \frac{1}{z - \Sigma(z)} \tag{14}$$

The problem of computing the Stieltjes transform is thus reduced to that of understanding which 1PI diagrams are dominating. We will see in the appendix that although the number of such diagrams is infinite, their contribution can be summed analytically, and a convenient elegant form for $\Sigma(z)$ can be derived.

## 4 RESULTS

In this section we present the main analytical predictions for the BBP thresholds and their comparison with finite-$N$ simulations. All derivations, including the field theory techniques used to compute the bulk distribution and outlier eigenvalue, are deferred to the Appendices A and B.

### 4.1 ANALYTICAL BBP TRANSITION

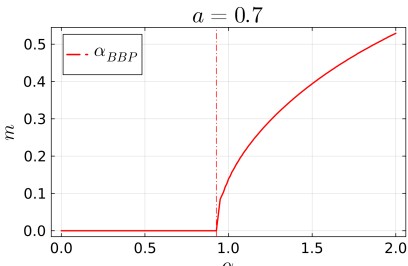 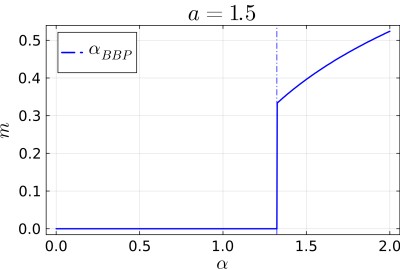

Figure 1: Overlap between the signal estimate and the true signal as a function of $\alpha$ for continuous BBP (*Left*) and discontinuous BBP (*Right*), with $p = 2$ and $p^* = 1$.

The critical value $\alpha_{\mathrm{BBP}}$ is analytically determined by imposing the condition

$$\lambda_*(\alpha_{BBP}) = \lambda_-(\alpha_{BBP}), \tag{15}$$

where $\lambda_-$ denotes the left edge of the bulk spectrum and $\lambda_*$ denotes the outlier eigenvalue. Depending on how the eigenvalue spectrum $\rho(\lambda)$ vanishes near its left edge when equation 15 is satisfied, the nature of the BBP transition can be one of two types, either **continuous** or **discontinuous** (Bocchi et al., 2026a; Bouchbinder et al., 2021; Potters & Bouchaud, 2020). A more detailed analysis of discontinuous BBP transitions and their finite size effects is addressed in Bocchi et al. (2026a). For the sake of completeness here we re-discuss some aspects in relation to their application to our teacher-student learning problem.

In the continuous BBP transition, the overlap $m$ between the eigenvector associated with the outlier eigenvalue and the signal(s) continuously grows from 0 to finite values as the signal-to-noise ratio $\alpha$ increases above the threshold $\alpha_{BBP}$. This case corresponds to a *sharp* edge of the spectrum, where the eigenvalue density vanishes with a square-root singularity:

$$\rho(\lambda) \underset{\lambda \to \lambda_-^{sh}}{\propto} (\lambda - \lambda_-^{sh})^{1/2}. \tag{16}$$

In contrast, if the BBP transition is discontinuous the value of the overlap immediately jumps from 0 to a finite value as soon as $\alpha > \alpha_{BBP}$. This occurs when the left edge of the spectrum is *smooth*, with the density decaying exponentially as

$$\rho(\lambda) \underset{\lambda \to \lambda_-^{sm}}{\propto} \exp\left[-\frac{A}{(\lambda - \lambda_-^{sm})}\right], \quad \text{for some constant } A > 0. \tag{17}$$

In Figure 1 we show the behavior of the overlap in the two different scenarios. While their difference is clear in the large dimensional limit $N \to \infty$, a distinction between the two types of transitions is also visible for finite system sizes, as discontinuous BBP transitions are characterized by a strong anticipation of the transition at $N$ finite, which we discuss in Section 4.2.

Depending on the student and teacher number of nodes $p, p^*$, as well as the normalizing constant $a$, either type of transition can occur. In what follows, we present results for $p^* = 1$, but we verify in the

appendix D that varying $p^*$ does not qualitatively alter the overall picture. Figure 2 shows the critical ratio $\alpha_{\text{BBP}}$, revealing two principal effects. First, $\alpha_{\text{BBP}}(a)$ shows non-monotonic dependence on $a$ at fixed $p$, with its minimum placed at a critical value $a_c(p)$ where the transition changes from continuous (left) to discontinuous (right). In other words, for given $p$ (and $p^*$), the most convenient $a$ allowing for an earliest recovery transition is the one where the BBP transition is at the verge of becoming discontinuous. Second, while increasing $p$ at fixed $a$ generally reduces $\alpha_{\text{BBP}}$, we again observe a critical threshold $p_c(a)$ beyond which the transition becomes discontinuous. Note that in its vicinity (see middle inset of Figure 2) a non-monotonic $\alpha_{\text{BBP}}(p)$ behavior is sometimes visible: the subsequent small increase of $\alpha_{\text{BBP}}(p)$ with $p$, *i.e.* increasing overparametrization, is at odds with the general expectation of the benefits of overparametrization in smoothening the landscape to let the signal emerge. However, as we will see in Section 4.2, this weak effect obtained in the infinite dimensional limit can be masked by strong, non-trivial finite-size effects, reinstating a general advantage of overparametrization for all practical purposes.

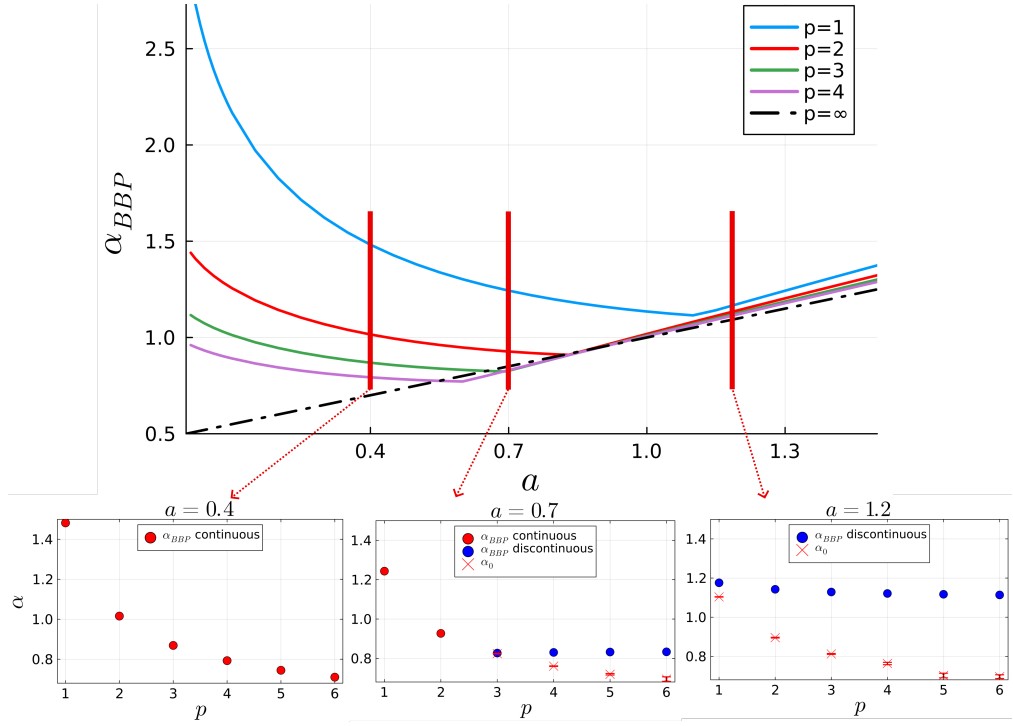

Figure 2: $\alpha_{BBP}$ as a function of $a$ for $p^* = 1$ and several values of $p$. The point at which the curves start increasing almost linearly is the point in which the transition becomes discontinuous. The dashed line shows $\alpha_{BBP}(a)$ in the large overparametrization limit where the transition is always discontinuous. The insets show $\alpha_{BBP}$ as a function of $p$ for three fixed values of $a$. Here, red points indicate the transition is continuous, while blue points that it is discontinuous. The red crosses are estimates of $\alpha_0$, a "finite-$N$" estimate of the transition described in section 4.2.

**Infinite overparametrization limit**   Before analyzing finite-size effects, we consider the limit of infinite overparametrization, $p \to \infty$. This limit is taken after the $N \to \infty$ limit, so we remain in the regime where $p \ll N$. In appendix B we show that in this limit the BBP transition is always discontinuous, with a threshold given by

$$\alpha_{\text{BBP}}^{p=\infty} = \frac{p^*(a+1)}{2}. \tag{18}$$

For any value of $p^*$, the minimum of $\alpha_{\text{BBP}}^{p=\infty}$ occurs at $a = 0$ and is equal to $p^*/2$, matching the information-theoretic weak recovery threshold identified in Maillard et al. (2024). Note that our setting is far from being Bayes optimal as in Maillard et al. (2024) since the overparametrized student, by definition, does not match the teacher structure. Yet, this result shows how powerful

overparametrization can be, as in the large overparametrization limit, even simply extracting spectral information from the Hessian at random configurations, the optimal weak recovery threshold can be achieved.

## 4.2 NUMERICAL SIMULATIONS AND FINITE $N$ BBP TRANSITION

In this section we compare the BBP thresholds calculated above with the empirical BBP threshold obtained in simulations of finite-dimensional problems with $p = p^* = 1$. For a fixed value of $a$, and over a range of values of $\alpha$, we generate the Hessian $\mathcal{H}$ for several values of the dimensionality $N$ of the problem, look at its eigenvalue spectrum, and count the number of times the eigenvalue associated to the eigenvector with maximum overlap with the signal is the smallest. In theory, this frequency, which we denote with the letter $\phi$, should go from 0 to 1 discontinuously in the $N \to \infty$ limit at $\alpha_{BBP}$. We perform this experiment for values of $a$ for which the transition is both continuous and discontinuous.

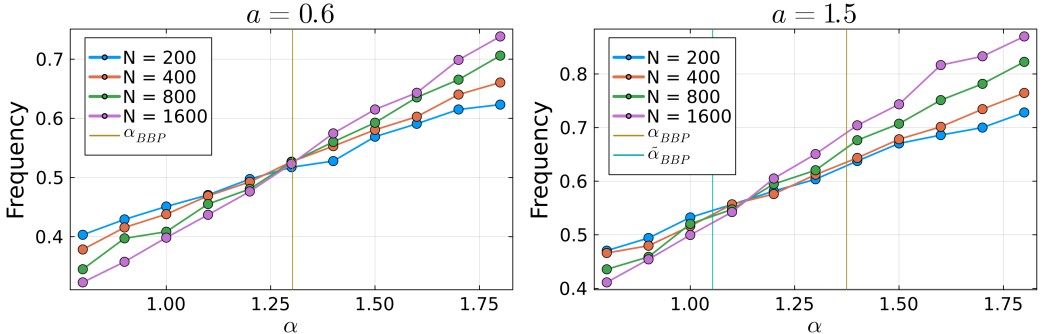

Figure 3: Comparison of BBP transitions for $p = p^* = 1$. On the $y$ axis we plot $\phi$, defined as the fraction of times the eigenvector with the maximum overlap with the signal corresponds to the smallest eigenvalue. On the left a value of $a$ for which the transition is continuous, on the right a value for which it is discontinuous. The vertical blue lines show our prediction for the BBP threshold, while for the discontinuous case the red line shows our estimate of $\alpha_0$.

We observe that when the transition is continuous, the predicted $\alpha_{BBP}$ threshold matches the point in which the curves for different values of $N$ intersect, while when it is discontinuous the threshold evaluated in the large $N$-limit is always above this point. That is, in the discontinuous case, our $N \to \infty$ prediction for $\alpha_{BBP}$ greatly overestimates the finite $N$ behavior. The explanation of this phenomenon is related to the shape of $\rho(\lambda)$ near the edge. When the transition is continuous the left edge is *sharp* and for a finite $N$ matrix then the typical deviation of the smallest eigenvalue of the bulk from the left edge is of the order of $N^{-2/3}$. When the BBP transition is discontinuous the left edge is *smooth* and, as a consequence, for a finite $N$ matrix it is much harder to sample this tail of the eigenvalue distribution and the smallest eigenvalue of the bulk will be larger than the $N \to \infty$ edge by a distance of the order of $1/\log(N) \gg N^{-2/3}$ (Bocchi et al., 2026a). Therefore the tails of the eigenvalue distributions for finite dimensional problems are much shorter than expected and allow the BBP eigenvalue to exit the bulk earlier. Unlike continuous BBP transitions, here the BBP eigenvalue retains a finite amount of information about the signal at the transition point and continues to do so for $\alpha < \alpha_{BBP}$, at least long as it remains the smallest eigenvalue of the finite-N matrices. These strong finite-size effects and the residual information explain why the observed algorithmic threshold (Figure 3, right panel) lies below the predicted $\alpha_{BBP}$ in the discontinuous case. To corroborate this explanation we conjecture [1] that the residual information $m$ about the teacher carried by the BBP eigenvalue for $\alpha < \alpha_{BBP}$ must decrease following a square root behavior until its vanishing at a smaller $\alpha_0$. Calculating the values of $m^2$ for various values of $\alpha > \alpha_{BBP}$, it is

---

[1] The conjecture is based on the fact that for smaller $N$ the domain where $g$ is not real, due to the singularity arising in the denominator of Equation equation 42, shrinks as $c_{min} < 0$ increases and $c_{max} > 0$ decreases. As a consequence it is possible to approach closer the point where $\frac{dz}{dg}$ vanishes and triggers the square root vanishing of $m$ as it is the case when approaching the continuous transition .

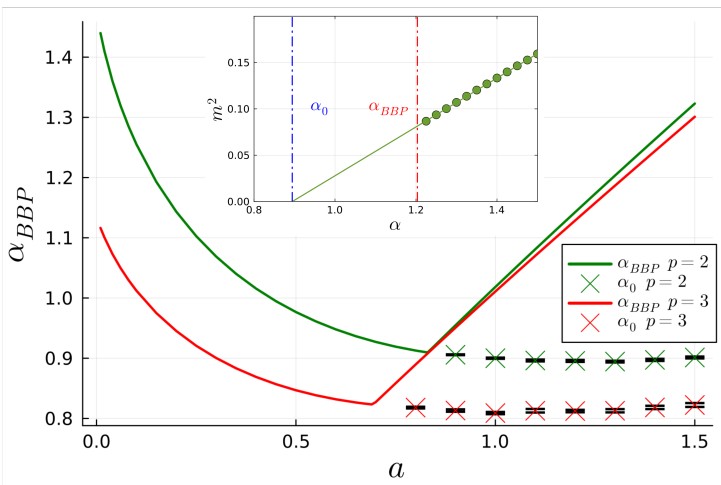

Figure 4: $\alpha_{BBP}$ (solid) and $\alpha_0$ (crosses) for two different values of $p$ as a function of $a$.

possible to perform a linear fit and obtain $\alpha_0$ from the intersection of its extrapolation to lower $\alpha$ with the $x$ axis. The result of this analysis is shown in Figure 4. The inset shows how the value of the threshold is extracted. On the right side of the plot we see the predicted discontinuous BBP transition as a function of $a$ for a couple of $p$ and $p^* = 1$ and the extrapolated signal to noise ratio $\alpha_0$ at which it is expected that the BBP eigenvalue will completely loose its information about the teacher. As we can see, this threshold lies below the corresponding predicted discontinuous BBP transition and slightly below the empirical transition for the finite dimensional version of the problem as shown in the right panel of Figure 3. The expectation is that $\alpha_0$ must represent a lower bound for the empirical transition at finite $N$ described above - as there is no hope that the lowest eigenvalue carries a finite correlation to signal when even the BBP eigenvalue has lost it - and that the empirical transition will very slowly move to the higher $\alpha$ as $N$ increases as the bulk eigenvalues will populate the tail progressively hiding the BBP eigenvalue. We also reported the estimated threshold $\alpha_0$ for the loss of information in the BBP eigenvalue in the middle and left insets of Figure 2. Note that it appears to always monotonically decrease with $p$, finally supporting the intuitive principle that overparametrization should favor learning.

## 5    CONCLUSION AND DISCUSSION

In this work, we presented a theoretical analysis of the loss landscape at initialization for a teacher-student setup with quadratic activation, considering networks with a generic, but finite, number of nodes, both for the teacher and the student. We investigated whether it is possible to extract information about the teacher simply by looking at the spectral properties of the Hessian at initialization, which reflects the curvature of the loss landscape in random configurations, without using iterative algorithms like gradient descent. In the high-dimensional data limit where both the input dimension $N$ and dataset size $M$ diverge while maintaining a finite signal-to-noise ratio $\alpha = M/N \sim O(1)$, we obtain that at small $\alpha$ the initial Hessian contains no information about the teacher, while at larger $\alpha$ one or more Hessian's principal eigenvectors develop a finite correlation with the teacher in a phenomenon called BBP transition. This approach resembles that of spectral algorithms (Mondelli & Montanari, 2018; Kovačević et al., 2025), which employ matrices that for some inference problems can be seen (Biroli et al., 2020) as Hessians averaged over many random choices of the student weights, to recover signals via spectral analysis. Nevertheless, our approach makes it possible to isolate the effect of overparametrization on this signal recovery transition. We complemented our theoretical findings with numerical simulations, fully characterizing this phenomenon for both finite and infinite $N$. Our analysis leads to the following key results:

**The BBP transition varies qualitatively with overparameterization and choice of loss.** Depending on the number of student nodes $p$ and the loss function's normalization constant $a$, the transition can be either continuous or discontinuous. The key difference between the two cases lies

in the overlap behavior at the transition: in the continuous case, the correlation with the teacher increases smoothly from zero when $\alpha$ increases, while in the discontinuous case, the outlier eigenvectors exhibit a finite overlap with the teacher immediately at the transition. Note that larger overparametrization is systematically associated to a discontinous BBP transition for signal recovery from the spectra of the Hessian at initialization. This result comes as the first practical application of the concept of a discontinuous BBP transition–very recently introduced and discussed in Potters & Bouchaud (2020); Bocchi et al. (2026a)–in association with overparametrization for a machine learning problem.

**Overparameterization tends to anticipate the transition, with notable exceptions.** Increasing $p$ (i.e., overparameterizing the student) for fixed $a$ generally lowers $\alpha_{\mathrm{BBP}}$, so that larger networks need less data to develop informative modes. Yet, for each fixed normalizing constant $a$, there exists a critical student size $p_c(a)$ beyond which the transition becomes discontinuous. Near this threshold, $\alpha_{\mathrm{BBP}}(p)$ can be non-monotonic and its precise shape depends on $a$. On the one hand, the overall trend confirms the generally established intuition that overparametrization is beneficial to learning, even extending it to the possibility to retrieve information about the teacher at initialization. On the other hand we observe notable, despite of small entity, exceptions to such behaviour. Surprisingly the entity of such exceptions ends up being further mitigated by finite size correction in empirical observations, reinstating a general advantage of overparametrization for most practical purposes.

**The large overparametrization limit achieves optimal performances.** In the limit of infinite overparametrization ($p \to \infty$), information about the teacher emerges through a discontinuous transition for all values of $a$. Intuitively, a highly overparameterized student can reproduce the teacher's weights an infinite number of times. This idea, discussed in other contexts (Biroli et al., 2020), helps explain why heavily overparameterized models can avoid overfitting and achieve better generalization, as if accessing an average view of the loss landscape. In our setting, the averaged Hessian at initialisation has a similar form to the spectral matrices used in (Mondelli & Montanari, 2018). However signal recovery via this spectral analysis, examined in (Bocchi et al., 2026a), does not quantitatively match the BBP transition in the large-$p$ limit, since it requires a stronger signal-to-noise ratio. Finally, as $a \to 0$, the BBP threshold $\alpha_{\mathrm{BBP}}$ in the large overparameterization limit converges to the information-theoretic threshold for weak recovery—a surprising result showing how strongly overparameterization can reshape the loss landscape to reveal the hidden signal. Remarkably, simple spectral analysis of the Hessian at initialization, far from Bayes-optimal conditions, suffices to match the weak recovery threshold in optimal settings.

**Finite-size correction affects the discontinuous BBP transition.** We compared the predictions for the BBP transition at different values of $p$, $p^*$, and $a$ with its numerical estimation for problems with finite-dimensional datasets, for several value of the dimensionality $N$. We obtained very good agreement in the case of continuous BBP transition but we observed a strong mismatch in the case of discontinuous BBP transitions. As also discussed in general in Bocchi et al. (2026a), we argued that these effects must be very strong–logarithmic in $N$. They are due to the smooth nature of the spectral edge, which finite-$N$ matrices fail to properly sample, and the large amount of residual information of the leading eigenvector even below the transition point. The undersampling of the tails gets stronger the lower $N$ and it allows the BBP eigenvalue to emerge earlier than the predicted BBP threshold, resulting in a numerical signal-recovery transition much lower than the predicted BBP transition. We also extract a lower bound $\alpha_0$ to the numerical transition evaluating the signal-to-noise ratio where the extrapolated overlap of the leading eigenvalue vanishes as a square root. The empirical transition is expected to slowly move from $\alpha_0$ to higher $\alpha$ approaching the BBP transition only in the large $N$ limit. Finally, surprisingly $\alpha_0$ is found to decrease with $p$ so that in the accessible finite-$N$ cases overparametrization turns out to be effectively advantageous for the empirical signal-recovery transition even when the predicted discontinuous BBP transition gets anomalously postponed. As suggested in Bonnaire et al. (2025), the fate of standard gradient descent should be influenced by the interplay between the emergence of the signal in the Hessian at initialization and the gradient-flow algorithmic transition. The latter can be predicted by evaluating the signal-to-noise ratio at which threshold states develop an instability toward the signal, also through a BBP transition. Understanding how overparameterization affects this second transition, both quantitatively and qualitatively, remains a very interesting open problem.

Finally, although our analysis focuses on the case of networks with quadratic activation functions, we show in appendix E that a qualitatively similar behavior is expected to hold in the more generic case with arbitrary activation functions.

**Acknowledgements** We thank Giulio Biroli for early discussions on the field theory approach to RMT, and Lenka Zdeborova for insightful exchanges. We acknowledge financial support from the European Union – NextGenerationEU (PNRR) Fund, Mission 4, Component 2. In particular, we acknowledge support from Investment 1.1 – Project 202234LKBW "Land(e)scapes: Statistical Physics Theory and Algorithms for Inference and Learning Problems", CUP B53D23003850006 (PRIN 2022). This work was also supported by Investment 1.3 – FAIR Foundation, Extended Partnership "Future Artificial Intelligence Research" (Project Code PE00000013-FAIR), and by Investment 1.4 – National Research Center in "High Performance Computing, Big Data and Quantum Computing" (Center Code CN00000013-CN1, ICSC).

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

**LLM Disclosure** LLMs were employed in this work to review the manuscript, correct grammatical errors, and polish the writing to improve clarity and readability.

## A  FIELD THEORY APPROACH

In this technical section, we give an overview of a field-theoretic approach (Zee, 1996; De Dominicis & Giardina, 2006) used to derive a self-consistent equation for the Stieltjes Transform of the Hessian, defined as

$$g(z) = \lim_{N \to \infty} \mathbb{E}_{\boldsymbol{x}} \frac{1}{Np} \operatorname{Tr} \boldsymbol{G}(z) = \lim_{N \to \infty} \mathbb{E}_{\boldsymbol{x}} \frac{1}{Np} \operatorname{Tr} \left( \frac{1}{z\mathbf{I} - \boldsymbol{\mathcal{H}}} \right). \tag{19}$$

The eigenvalue spectrum density can be obtained via the Stieltjes inversion formula:

$$\rho(\lambda) = \lim_{\epsilon \to 0^+} \frac{1}{\pi} \operatorname{Im} g(\lambda - i\epsilon). \tag{20}$$

Furthermore, we will build on the same formalism to obtain a self-consistent equation for the outlier eigenvalue $\lambda^*$, when it exists.

The first step is to use the rotational invariance of the teacher weight vectors. Without loss of generality, we fix them to lie along the first $p^*$ canonical directions $\boldsymbol{w}_{q_*}^* = \sqrt{N} \boldsymbol{e}_{q_*}$ for $q_* \in \{1, \ldots, p^*\}$, where the $\sqrt{N}$ ensures the correct normalization. With this choice, the teacher pre-activations reduce to $u_l^\mu = \sqrt{N} x_l^\mu \sim \mathcal{N}(0, 1)$. To separate the parts of the matrix where the coefficients $F_{qq'}$ are correlated to the components of the vectors $\boldsymbol{x}^\mu$, we permute the rows and columns of $\boldsymbol{\mathcal{H}}$ so that the first $p^*$ rows and columns of each block $\boldsymbol{\mathcal{H}}_{qq'}$ are grouped together in the top-left corner in a submatrix with elements $(\mathcal{H}^s)_{qq'}^{q_* q'_*} = \sum_\mu F_{qq'}^\mu x_{q_*}^\mu x_{q'_*}^\mu$ for $q, q' \in \{1, \ldots, p\}, q_*, q'_* \in \{1, \ldots, p^*\}$. $\boldsymbol{\mathcal{H}}^s$ should be understood as a block matrix, consisting of $(p^*)^2$ blocks, where each block is a $p \times p$ matrix $(\boldsymbol{\mathcal{H}}^s)^{q_* q'_*}$. The final shape of this permuted $\mathcal{H}$ is

$$\boldsymbol{\mathcal{H}} = \begin{pmatrix} \boldsymbol{\mathcal{H}}^s & \boldsymbol{\mathcal{H}}^c \\ (\boldsymbol{\mathcal{H}}^c)^T & \boldsymbol{\mathcal{H}}^b \end{pmatrix} \qquad \boldsymbol{\mathcal{H}}^s \in \mathbb{R}^{pp^* \times pp^*}, \boldsymbol{\mathcal{H}}^b \in \mathbb{R}^{(N-p^*)p \times (N-p^*)p}, \boldsymbol{\mathcal{H}}^c \in \mathbb{R}^{pp^* \times (N-p^*)p}. \tag{21}$$

In section A.1 we calculate the spectral distribution of its bulk eigenvalues $\rho(\lambda)$. It is a standard Random Matrix Theory result (see for example exercise 2.4.3 of Tao (2012)) that the spectral distribution does not change if we remove a number of rows and columns whose Frobenius norm is $o(N)$. Since every element of $\boldsymbol{\mathcal{H}}$ is $O(1)$, the Frobenius norm of $\boldsymbol{\mathcal{H}}^s$ is clearly $O(1)$, while by the strong law of large numbers the Frobenius norm of $\boldsymbol{\mathcal{H}}^c$ is $O(\sqrt{N})$. For the purpose of computing the spectrum bulk then we can simply discard them, and compute directly the Stieltjes transform of the matrix $\boldsymbol{\mathcal{H}}^b$. We will see that this leads to a great simplification.

In section A.2 instead we focus on the $pp^*$ outlier eigenvalues, which are not captured by the distribution $\rho(\lambda)$. In this case we cannot simply ignore the other blocks of the matrix $\mathcal{H}$. Indeed, if we divide the resolvent matrix $\boldsymbol{G}(z)$ in blocks in the same way,

$$\boldsymbol{G} = \begin{pmatrix} \tilde{\boldsymbol{G}} & \hat{\boldsymbol{G}} \\ \hat{\boldsymbol{G}}^T & \bar{\boldsymbol{G}} \end{pmatrix} \qquad \tilde{\boldsymbol{G}} \in \mathbb{R}^{pp^* \times pp^*}, \bar{\boldsymbol{G}} \in \mathbb{R}^{(N-p^*)p \times (N-p^*)p}, \hat{\boldsymbol{G}} \in \mathbb{R}^{pp^* \times (N-p^*)p}, \tag{22}$$

we have that the top left corner $\tilde{\boldsymbol{G}}$ encodes precisely for these outlier eigenvalues. Since $\boldsymbol{G}$ and $\mathcal{H}$ are related by an inverse, $\tilde{\boldsymbol{G}}$ will depend on all four blocks of $\mathcal{H}$. We will calculate it exactly using field theory.

### A.1 Spectrum Bulk

Following Zee (1996), the starting point for the calculation of the Stieltjes transform of $\mathcal{H}^b$ is to use a basic identity for Gaussian integration to write

$$g(z) = \lim_{N \to \infty} \mathbb{E}_{\boldsymbol{x}} \frac{1}{Np} \mathrm{Tr} \left( \frac{1}{z\boldsymbol{I}_{(N-p^*)p} - \mathcal{H}^b} \right) =$$

$$= \lim_{N \to \infty} \mathbb{E}_{\boldsymbol{x}} \frac{1}{\mathcal{Z}} \int \prod_{q=1}^{p} d\psi_q e^{-\frac{1}{2} \sum_{qq'} (\psi_q)^T \left( z\boldsymbol{I}_{(N-p^*)p} - \mathcal{H}^b \right)_{qq'} \psi_{q'}} \sum_q \frac{1}{Np} \|\psi_q\|^2. \tag{23}$$

Here, we introduced $p$ $(N - p^*)$-dimensional scalar fields $\psi_q$, and denoted by $\left( z\boldsymbol{I} - \mathcal{H}^b \right)_{qq'}$ the $qq'$-th block of the matrix $z\boldsymbol{I} - \mathcal{H}^b$. The next step is to get rid of the normalization constant $\frac{1}{\mathcal{Z}}$ using the replica trick $\frac{1}{\mathcal{Z}} = \lim_{n \to 0} \mathcal{Z}^{n-1}$ and introducing $n - 1$ replicas of the scalar fields $\{\psi_q^a\}_{a=1}^n$:

$$g(z) = \lim_{N \to \infty} \lim_{n \to 0} \mathbb{E}_{\boldsymbol{x}} \int \prod_{a=1}^{n} \prod_{q=1}^{p} d\psi_q^a e^{-\frac{1}{2} \sum_{aqq'} (\psi_q^a)^T \left( z\boldsymbol{I} - \mathcal{H}^b \right)_{qq'} \psi_{q'}^a} \sum_q \frac{1}{Np} \|\psi_q^1\|^2$$

$$\equiv \lim_{N \to \infty} \lim_{n \to 0} \left\langle \sum_q \frac{1}{Np} \|\psi_q^1\|^2 \right\rangle_{n,N}. \tag{24}$$

This integral cannot be performed analytically, as the $\boldsymbol{x}^\mu$ appear in the covariance matrix of the fields $\psi_q^a$. However, if we expand the exponential $e^{-\frac{1}{2} \sum_{aqq'} (\psi_q^a)^T \mathcal{H}^b_{qq'} \psi_{q'}^a}$, we are reduced to computing the average of every term with respect to the "bare" measure, namely the measure that appears in equation 24 with $\mathcal{H}^b$ set to zero. Since for every element of the Hessian in the bulk

$$(\mathcal{H}^b_{qq'})_{ij} = \sum_{\mu=1}^{\alpha N} F^\mu_{qq'} x^\mu_{p^*+i} x^\mu_{p^*+j}, \tag{25}$$

$F^\mu_{qq'}$ is independent of $x^\mu_{p^*+i}$, we can use Wick's probability theorem–according to which every higher order moment can be expressed as a function of second moments–to compute each term of the expansion. To wield the power of Feynman diagrams we identify two fields, one for $\psi_{iq}^a$ and one for $x_i^\mu$, which in accordance with Zee (1996) we call "quark" and "gluon" fields. Their bare propagators, $g^0$, are defined as the correlations of the fields in the "bare" measure. We will represent the former as straight lines and the latter as double lines.

$a, i, q$ ——————— $a, i, q$ $\qquad\qquad g^0_{quark} = \frac{1}{z}$,

$i, \mu$ ═══════════ $i, \mu$ $\qquad\qquad g^0_{gluon} = \mathbb{E}_{\boldsymbol{x}} x_i^2 = \frac{1}{N}$.

The interaction between the two fields can be read off $\mathcal{H}^b$, and can be represented with a vertex of the following kind and its corresponding weight

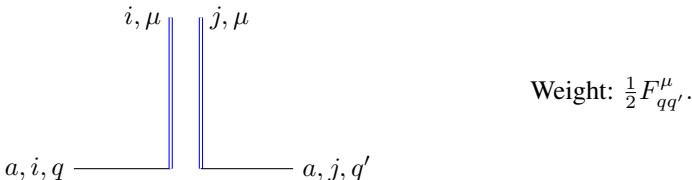

Weight: $\frac{1}{2} F_{qq'}^\mu$.

Note that although we are interested in the propagators $\langle (\psi_{iq}^1)^2 \rangle$, to derive a self-consistent equation we will have to consider also propagators between different blocks $\langle \psi_{iq}^1 \psi_{iq'}^1 \rangle$, which don't depend on the index $i$. From now on we will use $\boldsymbol{G}^b$ to indicate the $p \times p$ matrix formed by these elements, and with $\boldsymbol{G}_{quark}^0 = g_{quark}^0 \boldsymbol{I}_p$ the diagonal $p \times p$ matrix that contains the bare quark propagators on the diagonal. The Stieltjes Transform can be then obtained from $g(z) = \frac{1}{p} \text{Tr} \, \boldsymbol{G}^b(z)$.

Let us begin by examining the first set of diagrams. Since the contribution is identical for any index $i$, we may, without loss of generality, fix $i = 1$. In the diagrams shown below, propagators associated with $a, i = 1$ will be left unlabelled. When a propagator corresponds to a generic $i$ or $a$, we will explicitly annotate it on the line as a reminder that the corresponding index must be summed over. For each diagram, we will also indicate its total weight.
With one vertex we have diagrams:

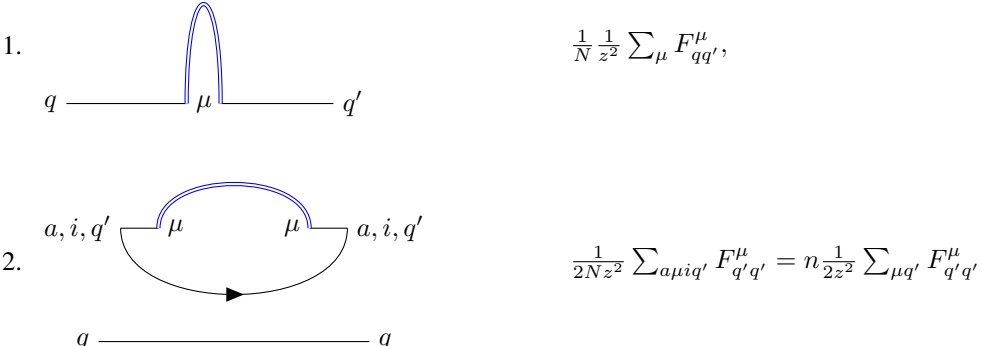

1. $\qquad \frac{1}{N} \frac{1}{z^2} \sum_\mu F_{qq'}^\mu$,

2. $\qquad \frac{1}{2Nz^2} \sum_{a\mu iq'} F_{q'q'}^\mu = n \frac{1}{2z^2} \sum_{\mu q'} F_{q'q'}^\mu$.

Note that the total contribution of diagrams of type 2 is proportional to the number of replicas, which in the limit $n \to 0$ goes to 0. In general, this holds for any disconnected diagram, so in the following we will focus on connected ones. Note also that in diagrams of type 1 the $1/2$ factor that comes from the weight of the vertex is canceled by a factor 2 that comes from the number of ways in which the vertex can be connected. Other than this $1/2$ factor, each diagram also carries an $1/n!$ factor, where $n$ is the number of vertices in the diagram, from the exponential expansion in equation 24. However, this is canceled by a factor $n!$ that comes from the number of ways the $n$ vertices can be aligned. This cancellation happens at all orders, so we will ignore such factors from now on.
Let us now consider two vertex diagrams:

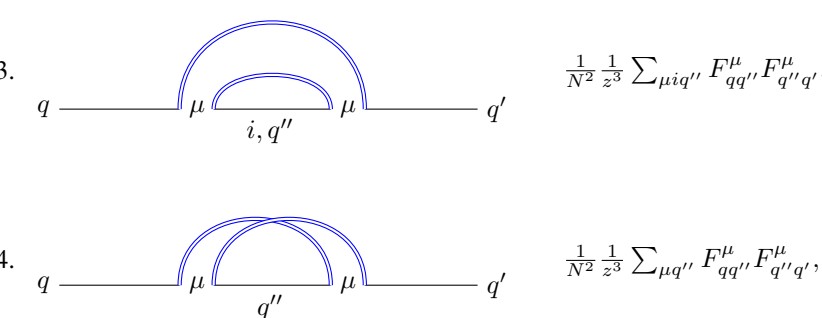

3. $\qquad \frac{1}{N^2} \frac{1}{z^3} \sum_{\mu iq''} F_{qq''}^\mu F_{q''q'}^\mu$,

4. $\qquad \frac{1}{N^2} \frac{1}{z^3} \sum_{\mu q''} F_{qq''}^\mu F_{q''q'}^\mu$,

5.

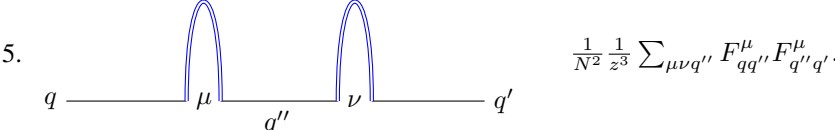

$$\frac{1}{N^2}\frac{1}{z^3}\sum_{\mu\nu q''}F^\mu_{qq''}F^\mu_{q''q'}.$$

The total weight of diagrams of type 4 is $o(1)$, so their contribution is negligible in the $N \to \infty$ limit. In general, this holds for all diagrams where gluon propagators intersect, so we will exclude these from now on.

Diagram 5 is instead obtained by connecting two diagrams of type 1 "in series". This is a general property of diagrammatic expansion: new diagrams can always be generated combining earlier ones in series through bare quark propagators. If we sum the contributions of all diagrams that are not obtained in this way, which in Quantum Field Theory are known as *one-particle irreducible* (1PI) diagrams, we can exploit this recursive structure to derive a self-consistent equation for the propagator $\boldsymbol{G}^b$. Let us call $\boldsymbol{\Sigma}^b$ the $p \times p$ matrix whose elements $\Sigma^b_{qq'}$ are the sum of all such 1PI diagrams connecting fields with block indices $q$ and $q'$, with external bare quark propagators removed (so-called *amputated* diagrams). Then we can express the sum of all diagrams as

$$\boldsymbol{G}^b = \boldsymbol{G}^0_{quark} + \boldsymbol{G}^0_{quark}\boldsymbol{\Sigma}^b\boldsymbol{G}^0_{quark} + \boldsymbol{G}^0_{quark}\boldsymbol{\Sigma}^b\boldsymbol{G}^0_{quark}\boldsymbol{\Sigma}^b\boldsymbol{G}^0_{quark} + \dots, \tag{26}$$

Factoring out $\boldsymbol{G}^0_{quark}$ reveals a geometric series, leading to

$$\boldsymbol{G}^b = \boldsymbol{G}^0_{quark}\left(\boldsymbol{I}_p + \boldsymbol{\Sigma}^b\boldsymbol{G}^0_{quark} + \boldsymbol{\Sigma}^b\boldsymbol{G}^0_{quark}\boldsymbol{\Sigma}^b\boldsymbol{G}^0_{quark} + \dots\right) = \left((g^0_{quark})^{-1}\boldsymbol{I}_p - \boldsymbol{\Sigma}^b\right)^{-1}, \tag{27}$$

This gives a self-consistent equation for the matrix $\boldsymbol{G}^b$, since $\boldsymbol{\Sigma}^b$ depends on $\boldsymbol{G}^b$, that in Physics is known as the *Dyson* equation. This allows us to focus on 1PI diagrams from now on.

Let us look at diagrams with 3 vertices.

6.

$$\frac{1}{N^3}\frac{1}{z^4}\sum_{\mu i j q''q'''}F^\mu_{qq''}F^\mu_{q''q'''}F^\mu_{q'''q'},$$

7.

$$\frac{1}{N^3}\frac{1}{z^4}\sum_{\mu\nu i j q''q'''}F^\mu_{qq''}F^\nu_{q''q'''}F^\mu_{q'''q'}.$$

As we can see, diagrams of type 7 are just diagrams of type 3 with a diagram of type 1 added to the inner quark propagator. This is a general property that is unique to this field theory: new diagrams can be obtained from previous ones by adding them to the inner quark propagators. If the starting diagram is 1-Particle Irreducible, then so is the new diagram. Since the sum of all possible diagrams is just the propagator, this form of combination can be accounted for by substituting $G^b_{qq'}$ to every inner bare quark line.

Indeed, the self-energy can be obtained from the sum of all leading order diagrams mentioned above, where the incoming and outgoing propagators are "amputated", and where every inner bare quark propagator is substituted by a propagator $G^b_{qq'}$, which we indicate graphically with a blob.

The first term in the sum is diagram of type 1, which gives a total contribution of $\frac{1}{N}\sum_\mu F^\mu_{qq'}$. The second term is given by diagram of type 3 with a $G^b_{qq'}$ propagator on its inner quark line

$$\frac{1}{N^2}\sum_{\mu i}\left(\boldsymbol{F}^\mu\boldsymbol{G}^b\boldsymbol{F}^\mu\right)_{qq'}.$$

Note that the incoming and outgoing propagators are shorter to indicate that we are considering amputated diagrams. The third term is diagram of type 6 where again we substitute propagators $G_{qq'}$

$$\frac{1}{N^3} \sum_{\mu i j} \left( \boldsymbol{F}^\mu \boldsymbol{G}^b \boldsymbol{F}^\mu \boldsymbol{G}^b \boldsymbol{F}^\mu \right)_{qq'}.$$

Note that we do not have to consider diagram of type 7 with propagators because it is already included in the second term. The next term is just the equivalent of the previous diagram but with three inner arches, from which we can guess the general form of the diagrams that appear in the self-energy.

Summing the weights of these diagrams, we can write in matrix form

$$\boldsymbol{\Sigma}^b = \frac{1}{N} \sum_\mu \boldsymbol{F}^\mu + \frac{1}{N} \sum_\mu \boldsymbol{F}^\mu \boldsymbol{G}^b \boldsymbol{F}^\mu + \frac{1}{N} \sum_\mu \boldsymbol{F}^\mu \boldsymbol{G}^b \boldsymbol{F}^\mu \boldsymbol{G}^b \boldsymbol{F}^\mu + \cdots =$$

$$= \frac{1}{N} \sum_\mu \boldsymbol{F}^\mu \left( \boldsymbol{I}_p + \boldsymbol{G}^b \boldsymbol{F}^\mu + \boldsymbol{G}^b \boldsymbol{F}^\mu \boldsymbol{G}^b \boldsymbol{F}^\mu + \dots \right) = \frac{P}{N} \frac{1}{P} \sum_\mu \boldsymbol{F}^\mu \left( \boldsymbol{I}_p - \boldsymbol{G}^b \boldsymbol{F}^\mu \right)^{-1} \rightarrow$$

$$\rightarrow \alpha \mathbb{E} \boldsymbol{F} \left( \boldsymbol{I}_p - \boldsymbol{G}^b \boldsymbol{F} \right)^{-1}, \tag{28}$$

where in the last identity we assumed that in the $N \rightarrow \infty$ the sum concentrates to its mean with respect to the dataset distribution.

Plugging this expression into equation 27, we get that the propagator satisfies the self-consistent equation

$$(\boldsymbol{G}^b)^{-1} = z \boldsymbol{I}_p - \alpha \mathbb{E} \boldsymbol{F} \left( \boldsymbol{I}_p - \boldsymbol{G}^b \boldsymbol{F} \right)^{-1}. \tag{29}$$

Note that the $\boldsymbol{F}$ matrix is of the form $\boldsymbol{F} = \alpha \boldsymbol{\lambda} \boldsymbol{\lambda}^T + \beta \boldsymbol{I}_p$ where $\boldsymbol{\lambda}$ is a standard Gaussian vector. In particular, it is rotationally invariant, so this equation should hold also for $\boldsymbol{O}^T \boldsymbol{F} \boldsymbol{O}$ where $\boldsymbol{O}$ is a rotation matrix.

$$(\boldsymbol{G}^b)^{-1} = z \boldsymbol{I}_p - \alpha \mathbb{E} \boldsymbol{O}^T \boldsymbol{F} \boldsymbol{O} \left( \boldsymbol{I}_p - \boldsymbol{G}^b \boldsymbol{O}^T \boldsymbol{F} \boldsymbol{O} \right)^{-1}, \tag{30}$$

from which

$$(\boldsymbol{O} \boldsymbol{G}^b \boldsymbol{O}^T)^{-1} = z \boldsymbol{I}_p - \alpha \mathbb{E} \boldsymbol{F} \left( \boldsymbol{I}_p - \boldsymbol{O} \boldsymbol{G}^b \boldsymbol{O}^T \boldsymbol{F} \right)^{-1}. \tag{31}$$

This must hold for any rotation matrix $\boldsymbol{O}$, so this implies that the matrix $\boldsymbol{G}$ must be proportional to the identity matrix. If we call $g(z)$ the value on the diagonal, equation 29 can be written in scalar form

$$g^{-1}(z) = z - \alpha \frac{1}{p} \sum_{l=1}^p \mathbb{E} \mathrm{Tr} \left[ \boldsymbol{F} \left( \boldsymbol{I}_p - g(z) \boldsymbol{F} \right)^{-1} \right] = z - \alpha \frac{1}{p} \sum_{l=1}^p \mathbb{E} \left[ \frac{c_l}{1 - g(z) c_l} \right], \tag{32}$$

where the $c_l$ are the eigenvalues of the $\boldsymbol{F}$ matrix.

## A.2 OUTLIER EIGENVALUE

The starting point for the calculation of the outlier eigenvalues is similar, with the exception that we have to consider the full matrix $\boldsymbol{\mathcal{H}}$ to calculate the elements of $\tilde{\boldsymbol{G}}(z)$:

$$\tilde{G}_{qq'}^{q_* q'_*}(z) = \lim_{n \to 0} \mathbb{E}_{\boldsymbol{x}} \int \prod_{a=1}^n \prod_{q=1}^p d\psi_q^a e^{-\frac{1}{2} \sum_a (\boldsymbol{\psi}^a)_<^T \left( z \boldsymbol{I}_{pp*} - \boldsymbol{\mathcal{H}}^s \right)(\boldsymbol{\psi}^a)_< + \sum_a (\boldsymbol{\psi}^a)_<^T \boldsymbol{\mathcal{H}}^c (\boldsymbol{\psi}^a)_>} \times$$

$$\times e^{-\frac{1}{2} \sum_a (\boldsymbol{\psi}^a)_>^T \left( z \boldsymbol{I}_{p(N-p*)} - \boldsymbol{\mathcal{H}}^b \right)(\boldsymbol{\psi}^a)_>} \psi_{q_* q}^1 \psi_{q'_* q'}^1. \tag{33}$$

Here, we introduce the notation $(\boldsymbol{\psi}^a)_<$ to denote for each $a$ the $pp^*$-dimensional vector with components $\psi^a_{iq}$ for $q \in \{1, \ldots, p\}$, $i \in \{1, \ldots, p^*\}$, and similarly $(\boldsymbol{\psi}^a)_>$ the $p(N - p^*)$-dimensional vector of components $\psi^a_{iq}$ for $q \in \{1, \ldots, p\}$, $i \in \{p^* + 1, \ldots, N\}$. The vectors are flattened in a way that the first $p$ components of $(\boldsymbol{\psi}^a)_<$ and $(\boldsymbol{\psi}^a)_>$ are respectively $\{\psi^a_{1q}\}^p_{q=1}$ and $\{\psi^a_{(p^*+1)q}\}^p_{q=1}$, the second $p$ components are $\{\psi^a_{2q}\}^p_{q=1}$ and $\{\psi^a_{(p^*+2)q}\}^p_{q=1}$ and so on.

As before, we can calculate this integral using diagrammatic expansion, with the only difference that we cannot average over the first $p^*$ components of the $\boldsymbol{x}^\mu$ field, since they appear in $F^\mu_{qq'}$ and we cannot simply use Wick's probability theorem. Indicating this time by a curly line the bare propagator between fields with indexes that belong to $(\boldsymbol{\psi}^a)_<$, and with straight lines indexes belonging to $(\boldsymbol{\psi}^a)_>$, we have that the bare propagators are

$$q, q_* \sim\!\!\sim\!\!\sim\!\!\sim\!\!\sim\!\!\sim\!\!\sim\!\!\sim q, q_* \qquad \tilde{g}^0 = \frac{1}{z - \frac{1}{N}\sum_\mu F^\mu_{\tilde{q}q}(u^\mu_{q_*})^2} \rightarrow \frac{1}{z - \alpha \mathbb{E} F_{qq} u^2_{q_*}}$$

$$a, i, q \text{ --------------- } a, i, q \qquad g^0_{quark} = \frac{1}{z}$$

$$i, \mu \text{ ============ } i, \mu \qquad g^0_{gluon} = \frac{1}{N}$$

The vertices are instead given by the old one

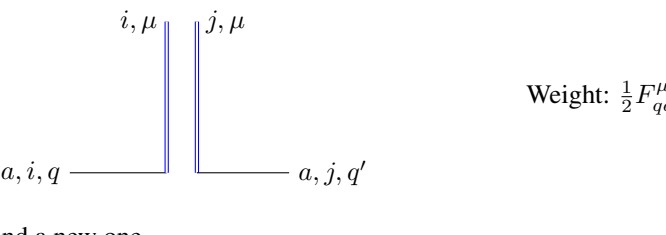

Weight: $\frac{1}{2}F^\mu_{qq'}$

and a new one

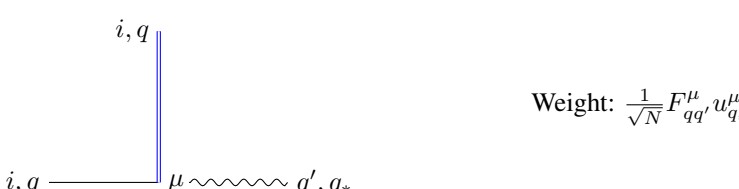

Weight: $\frac{1}{\sqrt{N}}F^\mu_{qq'}u^\mu_{q_*}$

As before, let us define the self-energy $\tilde{\Sigma}^{q_*q'_*}_{qq'}$ as the sum of all 1PI diagrams connecting the curly fields with indices $q, q_*$ and $q', q'_*$. Again interpreting $\tilde{\boldsymbol{\Sigma}}$ as a block matrix, with $(p^*)^2$ blocks of $p \times p$-dimensional matrices $\tilde{\boldsymbol{\Sigma}}^{q_*q'_*}$, then we can write a matrix Dyson equation for the resolvent

$$\tilde{\boldsymbol{G}} = \left( (\tilde{g}^0)^{-1} \boldsymbol{I}_{pp^*} - \tilde{\boldsymbol{\Sigma}} \right)^{-1}. \tag{34}$$

We expect the matrix $\tilde{\boldsymbol{G}}$ to be singular when $z$ is equal to the outlier eigenvalue, so a self-consistent equation can be obtained by imposing the singularity of $(\tilde{g}^0)^{-1}\boldsymbol{I}_p - \tilde{\boldsymbol{\Sigma}}$.

Let us now calculate the self-energy directly, using the same rules we found above, namely that disconnected and gluon-intersected diagrams are subleading. As we are looking for the self-energy of the "curly" field, we will necessarily need two vertices of the second kind.
The first term is given precisely by two such vertices

1. $q, q_* \sim\!\!\sim \mu$ ⊘ $\mu \sim\!\!\sim q', q'_*$ $\qquad \frac{1}{N^2}\sum_{\mu i}\left(\boldsymbol{F}^\mu \boldsymbol{G}^b \boldsymbol{F}^\mu\right)_{qq'} u^\mu_{q_*} u^\mu_{q'_*}$

$G^b_{q''q'''}$

Next, we can add a vertex of the first kind to get the diagrams

2. $q, q_* \sim\!\!\sim \mu \quad \mu \sim\!\!\sim q', q'_*$ $\qquad \frac{1}{N^3} \sum_{\mu i j} \left( \boldsymbol{F}^\mu \boldsymbol{G}^b \boldsymbol{F}^\mu \boldsymbol{G}^b \boldsymbol{F}^\mu \right)_{qq'} u^\mu_{q_*} u^\mu_{q'_*}$

3. $q, q_* \sim\!\!\sim \mu \quad \nu \quad \mu \sim\!\!\sim q', q'_*$ $\qquad \frac{1}{N^3} \sum_{\mu \nu i} \left( \boldsymbol{F}^\mu \boldsymbol{G}^b \boldsymbol{F}^\nu \boldsymbol{G}^b \boldsymbol{F}^\mu \right)_{qq'} u^\mu_{q_*} u^\mu_{q'_*}$

Diagrams of type 3 however are already counted in diagrams of type 1. We could also add two vertices of the second kind, but we would not obtain 1PI diagrams.

The next diagram is obtained by stacking one more arch to diagram 2, and again we can guess the iterative form of the diagrams that contribute. Summing the weight of all such diagrams, and remembering that $\boldsymbol{G}^b = g(z)\boldsymbol{I}_p$, we get that the self-energy is

$$\tilde{\boldsymbol{\Sigma}}^{q_* q'_*} = \frac{1}{N} \sum_\mu u^\mu_{q_*} u^\mu_{q'_*} \left( g(z) \boldsymbol{F}^\mu \boldsymbol{F}^\mu + g^2(z) \boldsymbol{F}^\mu \boldsymbol{F}^\mu \boldsymbol{F}^\mu + \dots \right) = \tag{35}$$

$$= \frac{1}{N} \sum_\mu u^\mu_{q_*} u^\mu_{q'_*} \left[ g(z) \left( \boldsymbol{F}^\mu \right)^2 \left( \boldsymbol{I}_p - \boldsymbol{F}^\mu g(z) \right)^{-1} \right] \to \alpha \mathbb{E} u_{q_*} u_{q'_*} \left[ g(z) \boldsymbol{F}^2 \left( \boldsymbol{I}_p - \boldsymbol{F} g(z) \right)^{-1} \right],$$

where again in the last equation we supposed that the sum concentrates to its mean with respect to the dataset.

Since $\boldsymbol{F}$ is an even function of $u_{q_*}$, $\tilde{\Sigma}^{q_* q'_*}_{qq'}$ is 0 for $q_* \neq q'_*$ (and in particular, the expectation will be the same for every index $q_*$). Using the fact that $\boldsymbol{G}^b$ is diagonal we can write

$$\tilde{\boldsymbol{\Sigma}}^{q_* q_*} = \alpha \mathbb{E} u^2_{q_*} \left[ g(z) \boldsymbol{F}^2 \left( \boldsymbol{I}_p - \boldsymbol{F} g(z) \right)^{-1} \right] \tag{36}$$

Again, $\boldsymbol{F}$ is rotationally invariant, so the same equation should hold for a rotated $\boldsymbol{F}$

$$\tilde{\boldsymbol{\Sigma}}^{q_* q_*} = \alpha \mathbb{E} u^2_{q_*} \left[ g(z) \boldsymbol{O}^T \boldsymbol{F}^2 \boldsymbol{O} \left( \boldsymbol{I}_p - \boldsymbol{O}^T \boldsymbol{F} \boldsymbol{O} g(z) \right)^{-1} \right] \tag{37}$$

from which

$$\boldsymbol{O} \tilde{\boldsymbol{\Sigma}}^{q_* q_*} \boldsymbol{O}^T = \alpha \mathbb{E} u^2_{q_*} \left[ g(z) \boldsymbol{F}^2 \left( \boldsymbol{I}_p - \boldsymbol{F} g(z) \right)^{-1} \right] \tag{38}$$

For this equation to hold for every rotation matrix $\boldsymbol{O}$, the matrix $\tilde{\boldsymbol{\Sigma}}^{q_* q_*}$ must be proportional to the identity. Consequently, we obtain $\tilde{\Sigma}^{q_* q_*}_{qq'} = \tilde{\Sigma}(z) \delta_{qq'} \delta_{q_* q'_*}$. Imposing the singularity of the RHS of equation 34, we get the equation

$$\left( \lambda^* - \alpha \mathbb{E} F_{11} u^2_1 - \tilde{\Sigma}(\lambda^*) \right)^{pp^*} = 0. \tag{39}$$

In other words, the $pp^*$ outliers all coincide with a value $\lambda^*$ which can be found by solving the self-consistent equation

$$\lambda^* = \alpha \mathbb{E} u^2_1 \frac{1}{p} \mathrm{Tr} \left[ \boldsymbol{F} \left( 1 + g(\lambda^*) \boldsymbol{F} \left( \boldsymbol{I}_p - \boldsymbol{F} g(\lambda^*) \right)^{-1} \right) \right] = \alpha \frac{1}{p} \mathbb{E} u^2_1 \mathrm{Tr} \left[ \boldsymbol{F} \left( \mathbb{I} - \boldsymbol{F} g(\lambda^*) \right)^{-1} \right]$$

$$= \alpha \mathbb{E} u^2_1 \frac{1}{p} \sum_{l=1}^p \frac{c_l}{1 - g(\lambda^*) c_l} \equiv \Xi(\lambda^*). \tag{40}$$

Combining this last expression with equation 32, one can rewrite a self-consistent equation for $g^* \equiv g(\lambda^*)$:

$$\frac{1}{g^*} = \alpha \mathbb{E} \left[ \frac{1}{p} \sum_{i=1}^p \frac{c_i (u^2_1 - 1)}{1 - g^* c_i} \right]. \tag{41}$$

# B   ANALYTICAL COMPUTATIONS FOR THE BBP

In the large dimensional limit, in order to obtain an expression for the left edge of the spectrum bulk we define the inverse function $z(g)$ of the resolvent $g(z)$, using equation 32, as

$$z(g) = \frac{1}{g} + \alpha\, \mathbb{E}\left[\frac{1}{p}\sum_{i=1}^{p}\frac{c_i}{1 - gc_i}\right]. \tag{42}$$

The density of eigenvalues $\rho(\lambda)$ of the spectrum bulk as a function of $z = \lambda$ will be obtained as the imaginary part of the $g$ solution to equation 42. In particular the edges correspond to the $z = \lambda$ where a non null imaginary part develops, which can occur by means of two different mechanisms. First: the domain of a $g$ real is constrained by the requirement that all denominators in equation 42 do not vanish. In particular equation 42 is well-defined for $g \in \mathbb{R} \setminus (\mathrm{supp}(1/c) \cup \{0\})$, where $\mathrm{supp}(1/c)$ denotes the support of $1/c$ across all $c_i$, *i.e.*, $\mathrm{supp}(1/c) = \bigcup_{i=1}^{p}\left\{\frac{1}{x} \mid x \in \mathrm{supp}(c_i)\right\}$. Assuming each $c_i$ has support within $[c_{i,\min}, c_{i,\max}]$, we define $c_{\min} = \min_i(c_{i,\min})$ and $c_{\max} = \max_i(c_{i,\max})$. If $c_{\min} < 0$ and $c_{\max} > 0$ (which will correspond to our case), then

$$g \in \left(\frac{1}{c_{\min}}, 0\right) \cup \left(0, \frac{1}{c_{\max}}\right) \equiv (g_{\min}, 0) \cup (0, g_{\max}). \tag{43}$$

Second: within this domain it can occur that $\frac{dz}{dg}$ vanishes. Beyond that point the solution to equation 42 cannot be real and a non zero imaginary part develops with a square-root behaviour. The second case corresponds to the standard square-root singularity at the edge of an eigenvalue distribution.

**Sharp edge**    If there exists a point $g_-^{sh} \in (g_{\min}, 0)$ at which the derivative of the inverse function

$$\frac{dz}{dg} = -\frac{1}{g^2} + \alpha\mathbb{E}\left[\frac{1}{p}\sum_{i=1}^{p}\frac{c_i^2}{(1 - gc_i)^2}\right] \tag{44}$$

vanishes, the left edge of the spectrum can be obtained by applying the inverse function to this value, i.e., $\lambda_-^{sh} \equiv z(g_-^{sh})$. The corresponding self-consistent condition is

$$g_-^{sh} = -\left\{\mathbb{E}\left[\frac{1}{p}\sum_{i=1}^{p}\frac{\alpha c_i^2}{(1 - g_-^{sh}c_i)^2}\right]\right\}^{-1/2}. \tag{45}$$

In this case, the left edge of the spectrum is *sharp*, meaning that the derivative of the eigenvalue density diverges as $\lambda$ approaches the edge from the right. This behavior arises through a standard square-root singularity at the edge:

$$\rho(\lambda) \underset{\lambda \to \lambda_-^{sh}}{\propto} (\lambda - \lambda_-^{sh})^{1/2}. \tag{46}$$

**Smooth edge**    If no solution $g_-^{sh} \in (g_{\min}, 0)$ exists such that the derivative of $z(g)$ vanishes, that is, if equation 45 cannot be satisfied within the domain of definition—then the left edge of the bulk spectrum is determined by the boundary of the domain itself. In this case, the edge of the resolvent is located at

$$g_-^{\mathrm{sm}} = g_{\min} = \frac{1}{c_{\min}}, \tag{47}$$

and the corresponding spectral edge is given by $\lambda_-^{sm} = z(g_-^{\mathrm{sm}})$. This edge is referred to as *smooth*, in the sense that the derivative of the spectral density $\rho(\lambda)$ vanishes as $\lambda \to \lambda_-^{sm}$. In our setting, this occurs via an exponential decay of the spectral density near the edge:

$$\rho(\lambda) \underset{\lambda \to \lambda_-^{sm}}{\propto} \exp\left[-\frac{A}{(\lambda - \lambda_-^{sm})}\right], \tag{48}$$

for some constant $A > 0$ (see Bocchi et al. (2026a) for more details about the derivation and the explicit form of the constants), indicating an essential singularity at the edge. In the case of our

teacher-student setup, the eigenvalues $c_i$ are

$$
\begin{cases}
c_1 = \dfrac{2}{p} \dfrac{1}{a + \frac{1}{p^*}\sum_{j=1}^{p^*} u_j^2} \left( \dfrac{3}{p} \sum_{i=1}^{p} \lambda_i^2 - \dfrac{1}{p^*} \sum_{j=1}^{p^*} u_j^2 \right) \\[4mm]
c_{2,\dots,p} = \dfrac{2}{p} \dfrac{1}{a + \frac{1}{p^*}\sum_{j=1}^{p^*} u_j^2} \left( \dfrac{1}{p} \sum_{i=1}^{p} \lambda_i^2 - \dfrac{1}{p^*} \sum_{j=1}^{p^*} u_j^2 \right),
\end{cases}
\tag{49}
$$

which have the same support $c_i \in (-2/p, +\infty)\ \forall i \in [1, p]$. This implies that for our setting $g_{min} = -p/2$. As we increase $\alpha$, for fixed values of $p$, $p^*$ and $a$, the left edge goes from being smooth to being sharp. That is, for $\alpha$ smaller than some value $\alpha_c$, the left edge is obtained from equation 47, while for larger $\alpha$ it is obtained from equation 45. The precise point in which this transition takes places, $\alpha_c$, can be determined by imposing the equation

$$
-\frac{p}{2} = \left\{ \mathbb{E}\left[ \frac{1}{p} \sum_{i=1}^{p} \frac{\alpha_c c_i^2}{(1 + \frac{p}{2} c_i)^2} \right] \right\}^{-1/2}.
\tag{50}
$$

Depending on the student and teacher number of nodes $p$, $p^*$, as well as the normalizing constant $a$, either type of transition can occur, as illustrated in Figure 5. In what follows, we present results for $p^* = 1$, but we verify in section D of the appendix that varying $p^*$ does not qualitatively alter the overall picture.

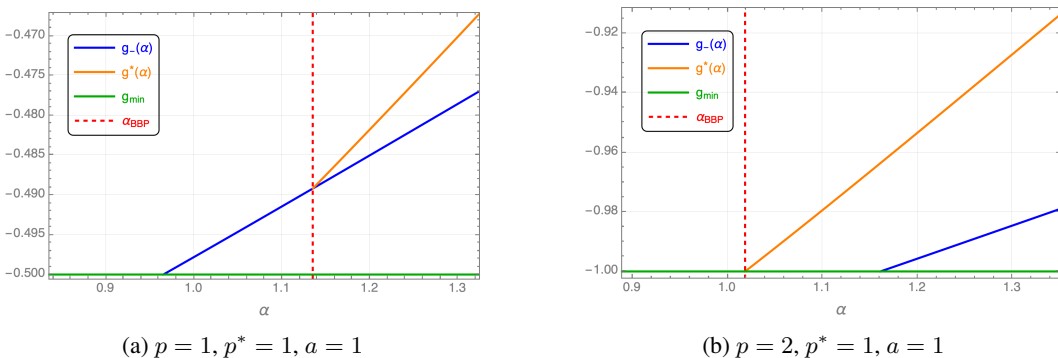

(a) $p = 1$, $p^* = 1$, $a = 1$         (b) $p = 2$, $p^* = 1$, $a = 1$

Figure 5: In Panel 5a, a continuous BBP is shown: the $\alpha_{\text{BBP}}$ corresponds to the crossing of the $g^*(\alpha)$ and $g_-(\alpha)$ curves. In Panel 5b, a discontinuous BBP is shown: the $\alpha_{\text{BBP}}$ corresponds to the crossing of the $g^*(\alpha)$ and $g_{\min}(\alpha)$ curves.

**Infinite overparametrization limit** In this paragraph we consider the limit of infinite over-parametrization, $p \to \infty$. This limit is taken after the $N \to \infty$ limit, so we remain in the regime where $p \ll N$. We begin by computing the critical value $\alpha_c$ below which the transition is discontinuous. For large $p$ we can approximate equation 50 as

$$
\frac{p}{2} \approx \left\{ \alpha_c \, \mathbb{E}\left[ \frac{4}{p^2} \cdot \frac{\left(1 - \frac{1}{p^*}\sum_l u_l^2\right)^2}{(1+a)^2} \right] \right\}^{-1/2},
\tag{51}
$$

which yields

$$
\alpha_c = \frac{p^*(a+1)^2}{2}.
\tag{52}
$$

We now turn to the BBP threshold $\alpha_{\text{BBP}}^{p=\infty}(a)$ in the $p \to \infty$ limit. This expression will be valid only for values of $a$ such that $\alpha_{\text{BBP}}^{p=\infty}(a) < \alpha_c(a)$. Evaluating equation 41 at $g^* = -p/2$, we obtain

$$
-\frac{p}{2} = \left\{ \alpha_{\text{BBP}}^{p=\infty} \, \mathbb{E}\left[ \frac{1}{p} \sum_{i=1}^{p} \frac{c_i(u_1^2 - 1)}{1 + \frac{p}{2} c_i} \right] \right\}^{-1},
\tag{53}
$$

which leads to the solution

$$\alpha_{\text{BBP}}^{p=\infty} = \frac{p^*(a+1)}{2}. \tag{54}$$

Note that for all $a > 0$, we have $\alpha_{\text{BBP}}^{p=\infty}(a) < \alpha_c(a)$, implying that the BBP transition is always discontinuous in the $p \to \infty$ limit. Furthermore, for general $p^*$, the minimum of $\alpha_{\text{BBP}}^{p=\infty}(a)$ occurs at $a = 0$ and is equal to $p^*/2$, matching the information-theoretic weak recovery threshold identified in Maillard et al. (2024). Note that our setting is far from being Bayes optimal setting as in Maillard et al. (2024) since the overparametrized student, by definition, does not match the teacher structure. Yet, this result shows how powerful overparametrization can be, as in the large overparametrization limit, even simply extracting spectral information from the Hessian at random configurations, the optimal weak recovery threshold is achieved.

## C   DERIVATION OF THE OVERLAP

In the general multi-index setting, each isolated eigenvalue may correspond to an alignment between a specific student node $\mathbf{w}_k$ and a teacher node $\mathbf{w}_l^*$. We denote the corresponding eigenvectors by $\mathbf{v}^{kl} \in \mathbb{R}^{pN}$ and reshape them into matrices $\boldsymbol{V}^{kl} \in \mathbb{R}^{p \times N}$, where the first $N$ entries of $\mathbf{v}^{kl}$ form the first row, and so on.

We define the overlap matrix:

$$\boldsymbol{M}^{kl} = \boldsymbol{V}^{kl}(\boldsymbol{W}^*)^\top \in \mathbb{R}^{p \times p^*}, \tag{55}$$

where $\boldsymbol{W}^* \in \mathbb{R}^{p^* \times N}$ is the matrix whose rows are the teacher weight vectors. Ideally, $\boldsymbol{M}^{kl}$ contains a single non-zero entry $m$ at position $(k,l)$, i.e., $(M^{kl})_{k'l'} = m_{kl}\delta_{k,k'}\delta_{l,l'}$.

However, the output of the student network in equation 2 can be equivalently written as

$$\hat{y}(\mathbf{x}^\mu) = (\mathbf{x}^\mu)^\top \frac{\boldsymbol{W}^\top \boldsymbol{W}}{p} \mathbf{x}^\mu, \tag{56}$$

where $\boldsymbol{W} \in \mathbb{R}^{p \times N}$ is the matrix whose rows are the student weight vectors. This expression reveals that the output is invariant under rotations of the matrix $\boldsymbol{W}$ in the $p$-dimensional space. That is, for any orthogonal matrix $\boldsymbol{O} \in \mathbb{R}^{p \times p}$, the transformation $\boldsymbol{W} \mapsto \boldsymbol{O}\boldsymbol{W}$ leaves the output function unchanged. A similar argument applies to the teacher network.

This rotational invariance implies that the student and teacher configurations are only identifiable up to orthogonal transformations, provided that the norms of the student nodes are unconstrained. As a consequence, the overlap matrix observed in practice takes the form

$$\boldsymbol{M}^{kl} = \boldsymbol{O}\boldsymbol{V}^{kl}(\boldsymbol{W}^*)^\top \tilde{\boldsymbol{O}}^\top, \tag{57}$$

where $\boldsymbol{O} \in \mathbb{R}^{p \times p}$ and $\tilde{\boldsymbol{O}} \in \mathbb{R}^{p^* \times p^*}$ are random orthogonal matrices[2]. The scalar overlap $m$ can then be extracted using the Frobenius norm:

$$m_{kl} = \sqrt{\sum_{i,j} \left(M_{ij}^{kl}\right)^2}. \tag{58}$$

Furthermore, due to the symmetry of the problem, all the overlaps will be equivalent:

$$m_{kl} = m \quad \text{for } k = 1, \ldots, p, \ l = 1, \ldots, p^*. \tag{59}$$

We now derive an analytic equation for $m$. Let us again choose the teacher vectors such that they are the first $p^*$ canonical directions, as we already did in the previous section. The starting point is to write the resolvent as

$$\boldsymbol{G}(z) = (z\boldsymbol{I}_{pN} - \boldsymbol{\mathcal{H}})^{-1} = \sum_{i=1}^{pN} \frac{\boldsymbol{v}_i \boldsymbol{v}_i^T}{z - \lambda_i} \tag{60}$$

---

[2]Note that such orthogonal transformations, while preserving the output, don't preserve the individual norms of the network nodes. If we apply this rotation to the teacher vector, which is taken to be normalized such that each node lies on the $S_{N-1}(\sqrt{N})$ sphere, such normalization property is lost. However, since the dataset is invariant under this rotation, the student cannot infer this normalization from the data, and will align to a randomly rotated teacher $\tilde{\boldsymbol{O}}\tilde{\boldsymbol{W}}^*$.

where $\{\lambda_i, \boldsymbol{v}_i\}_{i=1}^{pN}$ is the set of eigenvalues/eigenvectors. Without loss of generality, let us assume that we are in a base in which the first eigenvector is aligned with the first teacher. That is, if we call $\bar{\boldsymbol{e}}_1$ the first canonical direction in a $pN$ dimensional space, the overlap is $m = \boldsymbol{v}_1 \cdot \bar{\boldsymbol{e}}_1$. Multiplying equation 60 on the right and on the left by $\bar{\boldsymbol{e}}_1$, and taking the limit $\lambda \to \lambda^*$ we get that

$$\lim_{z\to\lambda^*} \underbrace{\bar{\boldsymbol{e}}_1^T \left(z\boldsymbol{I}_{pN} - \boldsymbol{\mathcal{H}}\right)^{-1} \bar{\boldsymbol{e}}_1}_{\tilde{G}_{11}^{11}(z)} = \lim_{z\to\lambda^*} \frac{m^2}{z - \lambda^*} \tag{61}$$

The square overlap is the residue of the function $\tilde{G}_{11}^{11}(z)$ at the pole $z = \lambda^*$. Using the relation found in the previous section $\tilde{G}_{11}^{11}(z) = (z - \Xi(z))^{-1}$, it can be calculated as

$$m^2 = \lim_{z\to\lambda^*} \frac{z - \lambda^*}{z - \Xi(z)} \tag{62}$$

where we remind the reader that

$$\Xi(\lambda) = \alpha \frac{1}{p} \sum_{i=1}^{p} \mathbb{E}u_1^2 \frac{c_i}{1 - g(\lambda)c_i}. \tag{63}$$

This limit gives the undetermined form $0/0$, but can be calculated using l'Hopital's rule

$$m^2 = \lim_{z\to\lambda^*} \frac{\partial_z(z - \lambda^*)}{\partial_z(z - \Xi(z))} = \frac{1}{1 - \partial_z\Xi(z)\big|_{z=\lambda^*}} \tag{64}$$

## D    DEPENDENCY ON $p^*$ AND UNDERPARAMETRIZATION

In this appendix we consider cases where $p^* \neq 1$. In figure 6 we show the equivalent of figure 2 for $p^* = 2$. As we can see the behaviour is qualitatively similar. For the values of $p$ we chose $\alpha_{BBP}$ is always monotonically decreasing as a function of $p$, although we don't expect this to hold for larger values of $p$.

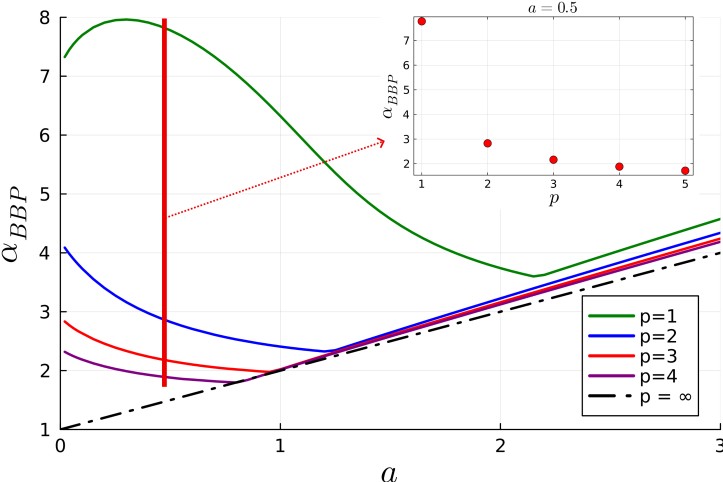

Figure 6: $\alpha_{BBP}$ as a function of $a$ for $p^* = 2$ for various values of $p$. In the inset we show $\alpha_{BBP}$ as a function of $p$ for a fixed value of $a = 0.5$

In figure 7 we explore the effect of underparametrization, by keeping $a$ and $p$ fixed and plotting $\alpha_{BBP}$ as a function of $p^*$. Perhaps unsurprisingly we observe that $\alpha_{BBP}$ increases, making the problem increasingly harder as the network is more underparametrized.

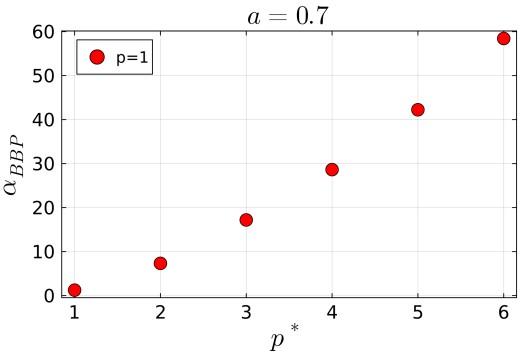

Figure 7: $\alpha_{BBP}$ as a function of $p^*$ for $p = 1$ and $a = 0.7$

## E    EXTENSION TO OTHER NON-LINEARITIES

In this section we consider the extension to arbitrary non-linearities of the learning task. We consider a generic non-linear activation function $\sigma(x)$. The teacher and student outputs become

$$\hat{y}(\mathbf{x}^\mu) = \frac{1}{p} \sum_{l=1}^{p} \sigma\left(\mathbf{w}_l \cdot \mathbf{x}^\mu\right) \equiv \frac{1}{p} \sum_{l=1}^{p} \sigma\left(u_l^\mu\right) \tag{65}$$

$$y(\mathbf{x}^\mu) = \frac{1}{p^*} \sum_{l=1}^{p^*} \sigma\left(\mathbf{w}_l^* \cdot \mathbf{x}^\mu\right) \equiv \frac{1}{p^*} \sum_{l=1}^{p^*} \sigma\left(\lambda_l^\mu\right), \tag{66}$$

The expression for the Hessian becomes

$$(\mathcal{H}_{qq'})_{ij} = \sum_{\mu=1}^{\alpha N} F_{qq'}^\mu x_i^\mu x_j^\mu \qquad F_{qq'}^\mu = \frac{1}{p} \sum_\mu \left[ \frac{\frac{1}{p}\sigma'(u_q^\mu)\sigma'(u_{q'}^\mu) + \delta_{qq'}(y^\mu - \hat{y}^\mu)\sigma''(u_q^\mu)}{a + y^\mu} \right], \tag{67}$$

Although repeating the theoretical derivation for such a matrix is beyond the scope of this work, let us comment that we expect a qualitatively similar behavior to the one outlined above. To support this claim, we show in figure 8 the overlap of the smallest eigenvector with the teacher $\mathbf{w}^*$ for two non-linear activation functions, $\sigma(x) = (1 + e^{-x})^{-1}$ the sigmoid function and $\sigma(x) = \tanh(x)$ the tanh function. In the both cases we choose $p^* = 1$ and $p = 3$. Let us note that, while the denominator in the loss function $(a + y)$ is a reasonable choice for a positive activation function, it is not for a function that can become equal to $-a$. In the case $\sigma(x) = \tanh(x)$ we choose a large value of $a$, such that the loss we are studying effectively reduces to the simple MSE loss. As we can see in both cases an outlier that is correlated with the teacher emerges, thus showing that this phenomenon is not restricted to the quadratic activation function.

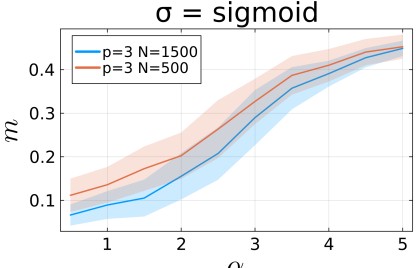
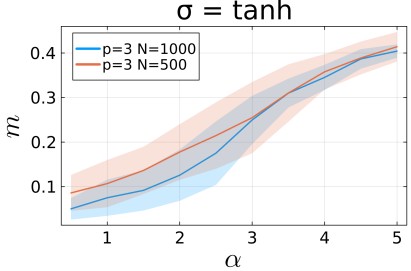

Figure 8: Overlap between smallest eigenvector of the Hessian and the true signal as a function of $\alpha$ for a sigmoid activation function (left) and for the tanh activation function (right). For the sigmoid activation $a = 1$, while to avoid the divergence of the denominator in the tanh case we set $a = 50$. Points are averaged over $n_{runs} = 60$ runs.

## F    CONNECTION TO DYNAMICS

To understand how the static BBP transition at initialization correlates with gradient descent dynamics, we performed simulations across different values of $p$, $N$, and $a$, keeping $p^* = 1$ for simplicity. We selected two values of $a$: $a = 0.1$, where the transition is continuous for all $p$, and $a = 2.0$, where it is discontinuous. Each simulation ran for 20,000 gradient descent steps, with the learning rate chosen so that the average initial update per student component was of order 0.1.

At initialization, the overlaps between student and teacher nodes, defined in equation 10, are all equivalent, and one could simply plot their average across all student-teacher pairs. However, the dynamics can break this symmetry and change student norms, consequently altering their relative importance. For example, condensation can occur: in the simple case of $p = 2$ and $p^* = 1$ one student could perfectly align with the teacher ($m_{11}^2 = 1$) and have order one norm, while the other has negligible norm and is uncorrelated ($m_{12}^2 = 0$). Such a network can achieve zero generalization error, yet a simple average of the magnetizations would yield $\langle m^2 \rangle = 0.5$. Since network performance is best captured by the most aligned student, we plot the maximum overlap as a proxy for final performance versus $\alpha$.

The results, shown in Figures 9 and 10, demonstrate that the BBP transition at initialization correlates with the point where dynamics begin to yield networks that perform better than random guesses, even in overparameterized settings. This earlier transition enables improved performance with fewer samples. The plots also suggest that as $N$ increases, the transition point shifts rightward, approaching the (currently unknown) BBP transition of the threshold states, which are the configurations that ultimately trap the dynamics for large $N$. Thus, while these preliminary results align with our intuition, a complete characterization of the BBP transition at the threshold states remains necessary to confirm this picture.

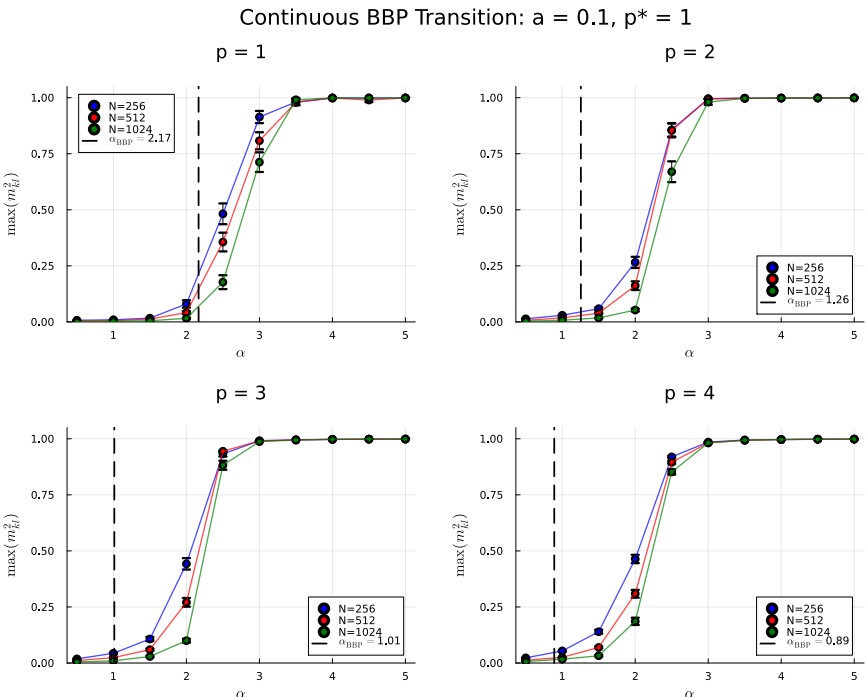

Figure 9: Maximum squared magnetization $m^2$ vs. $\alpha$ for $a = 0.1$. The BBP transition at intialization correlates with the onset of weak recovery.

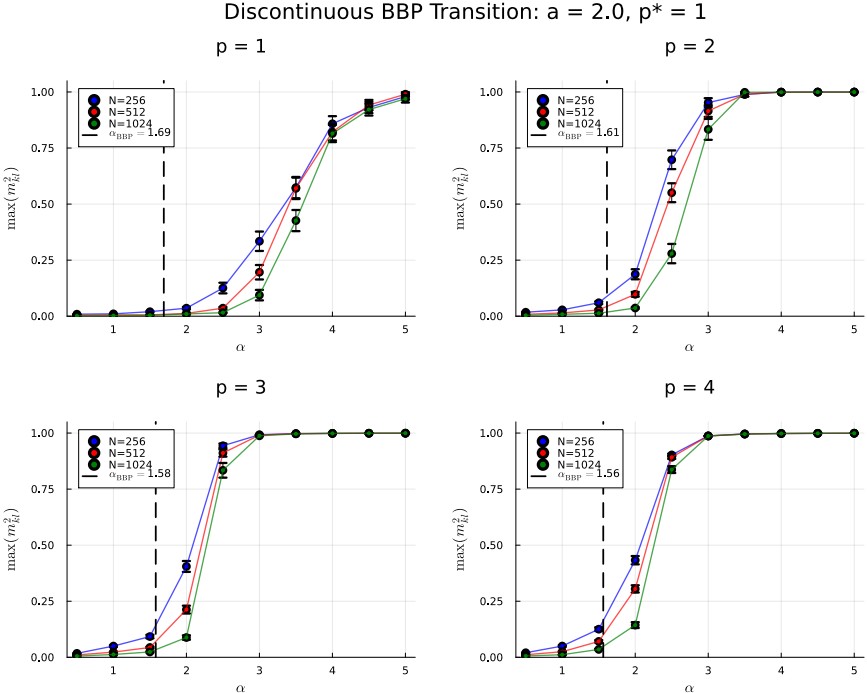

Figure 10: Maximum squared magnetization $m^2$ vs. $\alpha$ for $a = 2.0$. Also in the discontinuous case, the BBP transition correlates with the onset of weak recovery.

