# OpenReview forum: "Overparametrization bends the landscape: BBP transitions at initialization in simple Neural Networks"
_ICLR.cc/2026/Conference — ICLR 2026 Oral_

### Official Review · Reviewer_W5zG · 2025-10-29

**Soundness:** 3
**Presentation:** 3
**Contribution:** 3
**Rating:** 8
**Confidence:** 4

**Summary:**

The authors consider a teacher student model for finite-width, one-hidden-layer neural networks with quadratic activations (generalizing phase retrieval).
They study the Hessian of the loss landscape of the student at initialization, and characterize when a certain overlap between teacher weights and leading hessian eigenvector transitions from zero to finite (either in a continuous or discontinuous way). A positive overlap means that the Hessian at initialization contains easily accessible information on the teacher weights.
They compute the sample critical threshold, i.e. the minimum amount of samples M over dimension N such that there is information about the teacher weights in the initialization Hessian, and study the phenomenology as a function of the overparametrisation.

**Strengths:**

The authors quantify the effects of overparametrisation in a non-convex problem, in particular the effect at initialization. Given the context on spectral initialization and GD dynamics, I find it a nice way to discuss such effects.

The authors provide an application of the recently discussed discontinuous BBP transition, showing that such behavior is not a totally abstract curiosity.

**Weaknesses:**

The mismatch between finite N simulations of alpha_BBP in the discontinuous case and the theory should be characterized more precisely, even though it is clear that finite size corrections will be hard to "eliminate".
- Is there a different set of observables (other than that in Figure 3) that could be probed numerically to highlight the transition? Maybe with less important finite size effects?
- How prohibitive would it be to access finite size corrections from the field theory formalism?

**Questions:**

It is not apparent to me why one would need p,p* = O(1). What would fail in the derivation for p=O(d), for e.g.?

line 116: I would like to highlight also the following relevant paper https://arxiv.org/abs/2505.17958 where over-parametrized phase retrieval is considered also in an empirical minimization setting (finding the same weak recovery threshold as Maillard et al. 2024) in the complementary setting p^* = O(d), p=O(d) with p^*/d -> 0.

line 143: it is not clear to me why one would normalize the loss by 1/labels. Given that the authors remark that this is an important element of the subsequent analysis, it would be nice to have more intuition here.

line 157: is it "when" or "where"? If the first, when one would expect such Gaussianity to hold?

line 215: it would be nice to have an explicit definition of m here.

line 254-258: is there an intuition for why exponential decay at the bulk boundary induces a first order transition in the overlap?

line 333 and 351: is it the largest eigenvalue instead of smallest?

line 370 - 400: it is not clear to me how is alpha_0 computed: is it an analytical quantity? A numerical one? Also, it is not clear why it should be a lower bound to the finite-N transition. I suggest clarifying a bit.

---

> ### Author Response · Authors · 2025-11-20
> **Reply to Reviewer W5zG**
>
> ## Weaknesses
>
> > Is there a different set of observables (other than that in Figure 3) that could be probed numerically to highlight the transition?
>
> We agree that the characterization of finite-size ($N$) effects is not complete. However, this is a general challenge in the analysis of discontinuous BBP transitions and is not a problem specific to our model. A comprehensive theoretical characterization is therefore beyond the scope of this work. One might consider using the overlap $m$ between the student and teacher networks as an alternative observable, but for finite $N$, it suffers from similar issues: the transition appears continuous and only sharpens for very large $N$. To our knowledge, there is no known observable that fully mitigates these inherent finite-$N$ effects.
>
> > How prohibitive would it be to access finite size corrections from the field theory formalism?
>
>  While it is theoretically feasible to derive all diagrams contributing to the correction at a given order in $1/N$, their resummation is highly non-trivial. In this field-theoretic approach, obtaining an analytical result requires the exact resummation of an infinite number of diagrams; the theory is non-perturbative in the sense that diagrams with any number of vertices contribute at the same order. In our work, we have shown that this resummation can be performed exactly at the leading order, but a generalization to the $1/N$ corrections is not straightforward.
>
> ## Questions
>
> 1) Our analytical framework is based on reducing the problem of the spectrum of a $pN \times pN$ random matrix to two simpler problems: one involving large $N \times N$ random matrices and another involving a finite $p \times p$ matrix $\boldsymbol{F}$ (see Section A.1 of the appendix for details). In our case, the large $N \times N$ matrices are reweighted Wishart matrices, for which the continuous spectrum can be readily obtained. For the matrix $\boldsymbol{F} \in \mathbb{R}^{p \times p}$, if $p$ is $\mathcal{O}(1)$, one can derive the exact expressions for its $p$ eigenvalues and directly insert them into the general formula to obtain the result for the original matrix. If $p$ is not of order one, another layer of random matrix theory would be required to evaluate the now continuous spectrum of the $\boldsymbol{F}$ matrix, significantly complicating the analysis.
> As for $p^\*$, this work focuses on the realizable regime where it is of the same order as $p$ (specifically, $p^\* \leq p$), ensuring the student network has sufficient expressivity to match the teacher's labels.
>
> 2) We thank the reviewer for suggesting this relevant work. It has now been included in the literature review of our updated draft.
>
> 3) This is a common question among the reviewers, and for this reason we have dedicated a general comment to it. To briefly answer here: the intuition is that in the large $N$ limit, very rare events can cause the Hessian eigenspectrum to lack a finite left edge if the loss is not normalized. This means the eigenvalue spectrum for $N \to \infty$ spans to $-\infty$, which renders the appearance of an outlier to the left of the (non-existent) left edge impossible.
>
> 4) "Where" is indeed the appropriate term. In our setting, pre-activations are scalar products between two independent random vectors (the data and the student/teacher node) with components of finite variance. By the central limit theorem, these pre-activations converge to a Gaussian variable.
>
> 5) Yes we agree, we have added the explicit expression of $m$, alongside a brief sketch of the general analytical derivation in the updated version of the draft
>
> 6) In Appendix B we show that the main difference between the sharp edge case (square root behaviour at the boundary) and the smooth edge case (exponential decay at the boundary) manifests in the behavior of the inverse function of the Stieltjes transform, $z(g)$. Specifically, in the sharp edge case there exists a point where $z'(g)$ vanishes (or equivalently where $g'(z)$ diverges), while no such point exists in the smooth edge case. It can be shown that such derivative being zero or finite controls the overlap with the signal being zero or finite at the transition.
>
> 7) In our context, these are indeed the smallest (most negative or least positive, when they are all positive) eigenvalues. These eigenvalues of the Hessian matrix are associated with directions of the steepest negative curvature (or again least positive curvature).
>
> 8) In order to clarify the method used to obtain the value of $\alpha_0$ and its role as a lower bound, we have added a footnote at page 8 and added a sentence in the explanation in the main text. The parameter $\alpha_0$ is a heuristic measure. It is obtained by taking the analytical values of the squared overlaps for $\alpha$ above the transition, fitting a linear curve to them, and extrapolating this curve back below the transition (see Figure 4).

---

> > ### Comment · Reviewer_W5zG · 2025-11-24
> >
> > I thank the authors for engaging with my comments. I am fully satisfied by their answers, and I am happy to confirm both my rating and confidence scores.

---

### Official Review · Reviewer_x8Zk · 2025-10-30

**Soundness:** 3
**Presentation:** 2
**Contribution:** 3
**Rating:** 6
**Confidence:** 4

**Summary:**

The paper investigates how overparameterization affects the loss landscape at initialization in gradient-based learning. In particular, it considers a teacher-student setup where both teacher and student are two-layer neural networks with quadratic activations, generalizing previous works on phase retrieval to a multi-index setting. By analyzing the spectrum of the loss Hessian, the authors identify the conditions under which its leading eigenvector becomes informative about the teacher signal, and show that overparameterization bends the landscape, shifting the associated transition toward smaller sample sizes (lower SNR). In the infinite-width limit, the transition reaches the information-theoretic weak-recovery threshold. Finally, the authors investigate finite-size effects and compare them with the high-dimensional predictions.

**Strengths:**

The paper is theoretically and numerically sound. It addresses the important question of how overparameterization affects the learning landscape, offering novel, quantitative results in the specific setting of quadratic two-layer networks. Its originality lies in extending previous analyses of phase retrieval to a more general framework, providing a detailed characterization of the BBP phenomenology and valuable insights on finite-size effects. Overall, the presentation is clear and connects the findings to known information-theoretic thresholds and spectral methods.

**Weaknesses:**

1. One main weakness is the lack of a methodological overview in the main text. The technical analysis is confined to the appendix, leaving readers without intuition about the derivation. The paper could strongly benefit from a short but insightful methodological summary in the main section, possibly by shortening the (sometimes redundant) conclusion and/or using the additional page.

2. Some relevant references on the multi-index setting are missing. For instance, the critical SNR $p_\star/2$ has also been derived as the computationally optimal threshold in [1]; spectral method for multi-index models, included the quadratic network studied here, have been investigated and rigorously characterized in [2, 3].

3. The choice of the loss function may appear somewhat *ad hoc* and problem specific. It can benefit from further motivation and discussion.

[1] Troiani et al., "Fundamental limits of weak learnability in high-dimensional multi-index models"

[2] Kovačević et al., "Spectral Estimators for Multi-Index Models: Precise Asymptotics and Optimal Weak Recovery"

[3] Defilippis et al., "Optimal Spectral Transitions in High-Dimensional Multi-Index Models"

**Questions:**

1. Could you offer some intuition on how the label noise might qualitately affect the BBP phenomenology observed in this work?

2. In the conclusion, you mention that understanding the interplay between the emergence of a signal in the Hessian and the behavior of gradient-descent dynamics is an open direction. Do you have any preliminary numerical evidence or intuition on whether the BBP transition identified here corresponds to the point where gradient descent (or flow) begins to correlate with the teacher signal?
In particular, do you expect a qualitative behavior similar to the loss of correlation with the informative eigenvector observed in [Bonnaire et al., 2024], or would the overparameterized setting change this picture? Even a qualitative comment on whether such a connection is expected or not would be very interesting, although understandably beyond the main scope of this work.

---

> ### Author Response · Authors · 2025-11-20
> **Reply to Reviewer x8Zk**
>
> ## Weaknesses
>
> > One main weakness is the lack of a methodological overview in the main text.
>
> Indeed, the theoretical derivation was confined to the appendix to meet page limit requirements. We agree that a summary in the main text would be beneficial and have therefore updated the draft to include a compact methodological overview (see Section 3).
>
> > Some relevant references on the multi-index setting are missing.
>
> We thank the reviewer for bringing these closely related works to our attention. We have now integrated them into our literature review, with corresponding citations added in the introduction.
>
> > The choice of the loss function may appear somewhat ad hoc and problem specific.
>
> This is a common question between the reviewers and for this reason we have added a general comment on this. To briefly add a comment on this specific point, the loss acutally simply a mean square error loss (MSE) with the introduction of a denominator to cure instabilities due to very rare events, which doesn't qualitatively change the picture in practical applications.
>
> ## Questions
>
> > Could you offer some intuition on how the label noise might qualitately affect the BBP phenomenology observed in this work?
>
> Intuitively, label noise is expected to widen the bulk of the eigenvalue spectrum, thereby making the emergence of an outlier eigenvalue more difficult. This can be understood by analogy to the classical spiked Wigner model (Edwards, Samuel F. et al., *The eigenvalue spectrum of a large symmetric random matrix*, 1976), where the matrix undergoing the BBP transition is of the form $\sigma W + \alpha \boldsymbol{v} \boldsymbol{v}^T$. Here, $W$ is simply a Wigner matrix with entries $\mathcal{N}(0, 1/N)$ and $\boldsymbol{v}$ is the hidden signal. In this model, the effective signal-to-noise ratio is $\alpha/\sigma$, indicating that a larger noise magnitude $\sigma$ requires a stronger signal $\alpha$ to produce an outlier aligned with $\boldsymbol{v}$. We anticipate a similar mechanism is at play in our setting, where label noise effectively decreases the signal-to-noise ratio, thus raising the signal strength required for a distinct spectral outlier. Indeed, in the limit of infinite label noise strength, no outlier can emerge, as the labels become completely uncorrelated with the teacher.
>
> > Do you have any preliminary numerical evidence or intuition on whether the BBP transition identified here corresponds to the point where gradient descent (or flow) begins to correlate with the teacher signal?
>
> The connection between the initialization landscape and gradient descent dynamics is also a recurring question from the reviewers, and we have added a general comment to address it and a new section in the appendix of the updated draft (Section F).

---

### Official Review · Reviewer_upYX · 2025-11-01

**Soundness:** 4
**Presentation:** 4
**Contribution:** 3
**Rating:** 8
**Confidence:** 3

**Summary:**

This paper studies how overparameterization changes the geometry of simple neural networks at initialization.
The authors look at a two-layer quadratic network in a teacher–student setup, analyze the spectrum of the Hessian at random initialization, and show that it undergoes a Baik–Ben Arous–Péché (BBP) transition as the sample size (or SNR) increases.
The main finding is that wider networks “bend” the loss landscape: the BBP threshold shifts to smaller SNRs, the transition can become discontinuous, and in the large-width limit the threshold actually reaches the information-theoretic weak-recovery limit.
The work ties together ideas from random matrix theory, spectral initialization, and overparameterized learning, with clear analytical results and convincing numerical checks.

**Strengths:**

This is a strong theoretical contribution. The idea that overparameterization changes the nature of the BBP transition—and that in the large-width limit one reaches the optimal weak-recovery threshold—is both interesting and novel.It deepens our understanding of why wide models are easier to train, and it connects two previously separate lines of work: loss-landscape curvature and spectral initialization.

Original and timely topic: the interplay between overparameterization and loss-landscape geometry.
Technically clean derivations, connecting to known spectral and phase-retrieval results.
Solid numerical support and a nice discussion of finite-size effects.
Clear writing and good figures that make a rather technical story accessible.
Conceptually important: shows how widening a network effectively reshapes the curvature of the loss, anticipating information about the teacher signal even before training starts.

**Weaknesses:**

The analysis is limited to quadratic activations, which makes it less clear how general the conclusions are.

The field-theory derivations could be compressed; parts of the appendix are a bit heavy "physics-style"

It would have been nice to see a direct quantitative comparison with actual spectral initialization methods to highlight practical implications.

**Questions:**

Beyond quadratic activations:
The current analysis focuses on quadratic activations, which make the problem analytically tractable. It would be useful to discuss whether similar BBP behavior should appear for other smooth nonlinear activations (e.g., ReLU, erf). How much of the observed “bending of the landscape” is a consequence of the quadratic structure, and how much would persist in more realistic nonlinear settings where the Hessian couples to the input distribution?

Interpretation of discontinuous BBP transitions:
The discontinuous BBP transitions are a striking result. Can the authors clarify their physical or algorithmic interpretation? Do they correspond to a first-order–like instability in the optimization landscape, or to a sharp onset of alignment during early training dynamics?

Connection to optimization dynamics:
Since the paper analyzes the Hessian at random initialization, one may wonder why this spectral transition should meaningfully predict the onset of learnability for gradient descent in practice, especially at finite width. Are the BBP signatures expected to survive after a few optimization steps, or are they quickly “washed out” as the model moves in parameter space?

Robustness beyond Gaussian inputs:
How sensitive are the predicted thresholds to the Gaussian data assumption? Would structured or correlated inputs (e.g., non-isotropic covariance, nonzero mean) qualitatively alter the BBP critical point or the continuous/discontinuous nature of the transition?

Relation to implicit regularization and flatness:
Could the authors connect their findings to the broader literature on implicit regularization in overparameterized models—such as the bias of gradient flow toward flat minima? Does the observed “bending” of the Hessian spectrum have an analogue in the flatness or margin properties of trained solutions, or suggest a theoretical link between curvature at initialization and generalization in the final model?

---

> ### Author Response · Authors · 2025-11-20
> **Reply to Reviewer upYX**
>
> ## Weaknesses:
> > The analysis is limited to quadratic activations, which makes it less clear how general the conclusions are.
>
> Since this was a common question among reviews, we have addressed this issue in the general reply.
>
> > The field-theory derivations could be compressed; parts of the appendix are a bit heavy "physics-style"
>
> We believe that the field theoretic techniques outlined in the paper may be useful for many other computations in random matrix theory. We thus gave a very pedagogical overview of the technique, which can indeed seem at times "heavy", but we confined it in the Appendix. Given our objective, we believe that outlining all the details of the computation is a necessary first step for the popularization of the method, without altering the readability of the main text.
>
> >It would have been nice to see a direct quantitative comparison with actual spectral initialization methods to highlight practical implications.
>
> Compared to the optimal spectral method, we expect the Hessian to perform worse, in the sense that the BBP transition of the optimal spectral matrix is always below that of the Hessian, at least for finite $p$. While the former is built precisely to extract an alignment with the teacher for the least amount of data, the latter happens to have a similar form, but also has less freedom in the choice of the weighting prefactors $F^\mu$. It is therefore reasonable that it should not be optimal. For this reason, we don't expect that the use of the Hessian at initialization can have practical implications for problems where spectral methods have been already developed. On the other hand, a spectral analysis of the Hessian could be useful in problems where better spectral methods are not available.
>
> ## Questions
>
> > Beyond quadratic activations:
>
> This was a common question among reviewers, so we have dedicated a part of the general reply to it.
>
> > Interpretation of discontinuous BBP transitions:
>
> The discontinuous BBP is indeed a striking result. It is discontinuous in the sense that the alignement of the (BBP) eigenvector corresponding to the outlier eigenvalue should show a jump to a finite value when it exits the bulk spectrum. We don't expect a first-order-like instability in the landscape. Let us however comment that for common values of $N$ a bona fide jump is hardly observed as the BBP eigenvector keeps a finite correlation with the signal even when it enters the spectrum. For the finite $N$ case, where we expect the BBP at initialization phenomenon to be relevant for GD dynamics, it looses correlation with the signal only at the $\alpha_0$ threshold we define in our paper. For this reason at finite $N$, we expect the "bending" of the landscape to be qualitatively similar in the continuous and discontinuous cases, except that in the discontinuous case its prediction require awareness of the discontinous BBP phenomenology and of its finite size corrections.
>
> > Connection to optimization dynamics:
>
> Also this question was common among reviewers, so we dedicated a part of the general reply to the relation to optimization dynamics. For the case $p=p^\*=1$ , such relation was already outlined in Bonnaire et al. (2025). We expect a similar relation to hold in the overparametrised case, and we support this conjecture by showing that indeed overparametrization helps gradient descent in finding the signal $\boldsymbol{w}^*$. Let us stress that we expect the informative direction in the Hessian to help precisely in the finite $N$ case, as the time for correlation with the signal $m$ to become greater than zero during gradient descent is expected to scale as $t=O(\log N)$. In large-dimensional limit $N\to\infty$, such direction should be irrelevant, as the dynamics should follow the gradient and forget the initial condition.
>
> > Robustness beyond Gaussian inputs:
>
>
> Our results are only valid in the uncorrelated Gaussian inputs case. Beyond this settings, we cannot comment. We are aware of related work that studies spectral methods on correlated Gaussian inputs (Bousseyroux & Potters, 2024), and we expect the extension to the Hessian setting to be straightforward. This is an interesting open direction.
>
> > Relation to implicit regularization and flatness:
>
> This is another interesting question, which unfortunately we cannot answer with our setting. Understanding the flatness of the minimum reached by gradient descent would require studying another part of the landscape, namely the lowest part. This can be done in our setting by substituting the correct distribution $P(\boldsymbol{\lambda}, y)$ (which must be derived) to evaluate the averages $\frac{1}{p}\sum_{l=1}^p\mathbb{E} \frac{c_l}{1-g(z)c_l}$. We postpone this to future work.

---

### Official Review · Reviewer_Z5EM · 2025-11-10

**Soundness:** 3
**Presentation:** 2
**Contribution:** 3
**Rating:** 4
**Confidence:** 2

**Summary:**

This paper discusses a phase transitional behavior of the Hessian at initialization in the overparametrized setting. They discuss the setup where the teacher and student networks are two-layer neural networks with quadratic activation, and the width of the student is possibly larger than the width of the teacher. In classical BBP transition analysis, it is known that there is a threshold where the SNR is larger than this threshold the largest eigenvector aligns with the true signal. They analyze the threshold in the overparametrized student-teacher setup, and show that 1) transition happens either continuously or discontinuously 2) overparametrization decreases the threshold and makes training easier 3) as the width of the student -> infinity, the threshold becomes optimal. The claims are supported with experiments.

**Strengths:**

1. An interesting perspective on neural network training by discussing overparametrized phase retrieval - the theory involves machinery from quantum mechanics (of which I did not fully understand), which shows an interesting link between learning theory and physics.

2. The related works are cited extensively and mentioned appropriately in relevant parts of the manuscript.

**Weaknesses:**

1. Clarification in terminology is needed.
 - Why would this be a "loss landscape" result? Seems to me that the result is mostly on Hessian "at initialization" - which to me, it is not natural to understand the result as loss landscape result (of course, Hessians and loss landscape are related, but the training dynamics is not discussed).
 - What does "bend the landscape" mean?
 - What is the SNR that is repeated throughout the paper? I assume it would be alpha = M/N, I am wondering if the terminology SNR is used to express such quantity.
 - Using the term overparametrized could be a little bit misleading, because in general the term is used to state that the number of parameters >= number of data points.

2. It is a little hard to understand the technical novelty of the paper. What is the technique that is needed to study the overparametrized setting, which is different from previous approaches? Are there any challenges?

3. The experiments are good in the sense that the discoveries are verified, but the experiments are quite small-scale. $p, p^{*}$ are in the scale of 1,2,3,4. Larger experiments may be helpful. Also, I can see that the threshold becomes smaller when overparametrization occurs, but does that imply that it needs to better training? I don't see direct evidence of it. For instance, it would be helpful if there is an experiment where you apply gradient descent on the actual learning problem and show that overparametrization yields faster convergence/better generalization etc.

4. It would be good if we could see justifications of certain theoretical problem settings. e.g. Why should the activation be quadratic? Why should we train with normalized quadratic loss function?

**Questions:**

See weaknesses.

---

> ### Author Response · Authors · 2025-11-20
> **Reply to Reviewer Z5EM**
>
> ## Weaknesses and Questions
>
> > Why would this be a "loss landscape" result?
>
> Indeed in this work we focus on a particular part of the landscape, namely the high loss points that are sampled by a randomly initialised neural network. In Bonnaire et al. (2025) the authors showed that, perhaps counterintuitively, the geometry of this part of the landscape can have a strong effect on the gradient descent dynamics, at least for finite $N$. This motivates us to study the Hessian at initialisation, which already displays a rich phenomenology. However, to understand the full picture, a detailed study of the bottom of the landscape, and how it can change with overparametrization, will be necessary. We postpone this to future work.
>
> > What does "bend the landscape" mean?
>
> The verb "bends" is used in a pictorial sense to describe the change in the curvature of the landscape: less data is required for a further downward bending (or at least a reduced upward bending, if curvatures are all positive) to appear in the direction of the teacher.
>
> > What is the SNR that is repeated throughout the paper?
>
> Indeed we did not explicitly state that the SNR described in the introduction is the $\alpha$ parameter introduced in the second section. We thank the reviewer for pointing this out. We have now made the link explicit at the beginning of section 2.
>
> > Using the term overparametrized could be a little bit misleading,...
>
> This is a subtle point. It is true that overparametrization is sometimes measured as the ratio between the number of parameters and the size of the dataset, which in our notation would be $1/\alpha$. Another possible way of defining it would be the ratio between the number of parameters and some measure of the hardness of the task. In a teacher student setting, this hardness is better captured by the number of parameters of the teacher network rather than the number of inputs. It is thus pretty common to define overparametrization as the ratio between the number of parameters between the student and teacher, in our notation $p/p^*$. Intuitively, it also makes sense to define an overparametrized network as a network that has enough parameters to the fit the data, no matter how big the size of the dataset.
>
> > It is a little hard to understand the technical novelty of the paper...
>
> The field-theoretic approach to studying random matrices is not new, as it was first introduced nearly thirty years ago in (Zee, 1996). However, despite its long history, it has been rarely applied within the Statistical Physics and Theoretical Machine Learning communities. In this work, we employ this technique to compute two key quantities: the spectrum of the matrix and the outlier. The former can be obtained through various alternative methods (replica method, free probability theory,ecc...) and was already derived using field theory in the original works of the 1990s. The latter, by contrast, has not previously been computed using a field-theoretic approach. To the best of our knowledge, for this task, simple and direct methods such as the one we apply here have never been used before. Finally, the straightforwardness of the field-theoretic approach is crucial for the derivation of the outlier in the  overparametrized case where a more structured Hessian matrix must be handled.
>
> > The experiments are good in the sense that the discoveries are verified,...
>
> The size of our simulations is limited by the fact that the size of the Hessian is $pN\times pN$. Already when $p=5$ and $N=2000$ the matrix is large enough that diagonalization becomes computationally expensive. Also for the analytical part, obtaining the theoretical curves requires solving a fixed point iteration where each iteration requires a numerical integration of a $p$-dimensional integral. For $p>6$ this starts to become expensive, so effectively we are limited in the number of nodes we can choose. Also we show what happens in the limit $p\rightarrow \infty$ when the analytic equations simplify. Finally, we focused on this regime of small $p$ on purpose to address the question about the effect of overparametrization on the persistence of a hard phase for gradient descent. In that, our study is a first step towards filling the gap between the large wealth of papers on Phase Retrieval and a few interesting recent papers focusing on the large $p>N$ where the landscape has become smooth, e.g. (Sarao Mannelli et al, 2020). As for the relation between lower BBP threshold and better training, this is addressed in the official general reply.
>
> >It would be good if we could see justifications of certain theoretical problem settings...
>
> Since these questions were common among reviews, we have written a general reply addressing them.

---

### Author Response · Authors · 2025-11-20
**General Reply**

First of all we deeply thank all the referees for the many insightful suggestions and stimulating questions.
We addressed the specific points in the individual answers and we devote this general reply to topics raised by more than one reviewer.

## Quadratic Activations

The reason we focus on the quadratic activation function is that it can be connected to the Phase Retrieval problem, which is an interesting well studied learning task for which a huge wealth of results exists. In (Bonnaire et al., 2025), the authors observe a BBP-at-initialisation phenomenon precisely for this activation function. Furthermore, particular features of this problem make the study of the Hessian at initialisation particularly interesting.  For example, we expect this informative eigenvector to be particularly relevant for non-convex rough landscapes, where the optimization task with gradient descent is known to be hard. By choosing the quadratic activation we are ensuring that we are thus in such a relevant setting.
Our derivation can be repeated for a generic non-linear activation function $\sigma(x)$. The slight complication is that the matrix $\boldsymbol{F}^\mu = a\,\sigma'(\boldsymbol{\lambda})\sigma'(\boldsymbol{\lambda})^T + b\,\text{Diag}\big(\sigma''(\boldsymbol{\lambda})\big)$ is not rotationally invariant anymore, and does not allow explicit analytic expressions for its eigenvalues for $p$ larger than 3. Both problems can be overcome by writing the fixed point equation for the full matrix $\boldsymbol{G}^b$, and evaluating averages such as $\mathbb{E}\boldsymbol{F}\left(\boldsymbol{I}_p-\boldsymbol{G}^b \boldsymbol{F}\right)^{-1}$ using Monte Carlo methods.
We would like to stress that although the theoretical analysis is more complicated, the same theoretical picture is expected to hold for different nonlinearities. We have now added a section in the appendix where we show that a similar BBP-like phenomenon holds for Hessian defined with the sigmoid and tanh activation functions. In both cases an outlier that is correlated with the teacher emerges to the left of the spectrum of the Hessian at initialization, thus showing that this phenomenon is not restricted to the quadratic activation function. Preliminary numerical simulations seem also to show the advantage of overparametrization from this point of view.

## Choice of Loss Function

All analytical computations performed in this work can be applied to any loss function that provides a finite left edge in the Hessian eigenspectrum. The specific loss function used here is a generalization of the classical mean squared error (MSE) which incorporates a denominator $(y + a)$. The need of the denominator is due to the unboundedness of $y$. Numerically, this term acts as a regularizer to prevent instabilities arising from rare events with very small or large teacher outputs $y$.   Analytically, its purpose is to condition the Hessian eigenspectrum, guaranteeing a finite left edge in the $N \to \infty$ limit. A bounded activation function should not require any additional regularizer term in the loss to grant the existence of a finite left edge in the Hessian spectrum.
Consequently, under these conditions, while the precise transition points may vary with the choice of loss function, the qualitative mechanisms and interpretations proposed in this work are not unique to this specific form and are expected to generalize to other regression losses.
If one instead chooses a combination of activation function and loss function that does not show a finite left edge, there is currently no known theory that can obtain these kinds of results for $N \to \infty$, even if numerical simulations suggest that the qualitative finite size behavior remains similar.

## Connection to Dynamics

We thank the reviewers for encouraging us to explore the connection between static phase transitions and learning dynamics. In response, we have added a new Appendix F presenting preliminary results on how the BBP transition at initialization correlates with gradient descent dynamics.  Our simulations across different values of $p$, $N$, and $a$ (with $p^* = 1$) reveal that the BBP transition marks the onset of weak recovery in the dynamics, with overparameterization generally anticipating this transition.
From the results of (Bonnaire et al, 2025) we expect gradient descent dynamics at finite $N$ to be controlled not only from the BBP transition in the Hessian at initialization, but also from the BBP transition of the Hessian at threshold states at higher SNR, which should remain the only discriminant in the large $N$ limit.
These preliminary results indeed show that the curves move to the right as $N$ increases, aligning with the intuition that as $N$ increases, the dynamical transition will shift toward a later BBP transition of threshold states.
A complete characterization of this latter transition remains an important direction for future work.

---

### Author Response · Authors · 2025-12-03
**Quick Summary for the Area Chair**

We have engaged in a discussion with all four reviewers. We have addressed some specific questions in the reply to each referee, and common concerns in the general reply. We have updated the draft accordingly, in particular adding a section in the main text, as suggested by one of the referees, and two sections in the appendix, addressing these common concerns.
The only reviewer we received a reply from was satisfied by our answers, and confirmed his/her positive rating.
We believe the updated draft is significantly improved, and thank all the reviewers for their suggestions.

---

### Meta-Review · Area_Chair_A6GW · 2026-01-01

**Summary:**

This paper investigates the impact of overparametrisation on the loss landscape of two-layer neural networks using a teacher-student framework with quadratic activations. By applying a combination of mean-field and random matrix theory techniques to analyse the Hessian spectrum at initialisation, the authors identify a BBP transition that demarcates informative from uninformative regimes. The findings show that overparametrization effectively *"bends"* the landscape, shifting the transition to lower signal-to-noise ratios and allowing the system to reach the information-theoretic weak-recovery threshold in the infinite-width limit. The work further distinguishes between continuous and discontinuous transitions and provides estimates for signal recovery thresholds in finite-size settings.

The primary concerns focused on clarity, the generalisability of the framework, and its connection to practical optimisation.
Reviewers questioned the choice of quadratic activations and Gaussian inputs, asking whether the phenomenon persists with other non-linearities like sigmoid or ReLU. Another comment concerns the choice of a normalised quadratic loss function, which appears problem-specific. Another critique involved the difference between the static analysis of the Hessian at initialisation and training dynamics, in particular the improved convergence of GD. Finally, clarity was also criticised, asking to add definitions and a methodological introduction. These concerns were addressed in the rebuttal by adding a section in the main and appendices on non-linear activations and dynamic connections,

In conclusion, the main concerns have been addressed and I **recommend this paper to be accepted as oral**.

**Reviewer Concerns:**

- **Generalisability:** Several reviewers asked about different possible generalisations, in particular: activation function (Reviewer upYX and Z5EM), loss function (Reviewers Z5EM, x8Zk, and W5zG), and Guassian inputs (Reviewer upYX). *Rebuttal:*  The authors added an appendix showing similar BBP phenomena for sigmoid and tanh activations, and clarified that the normalised loss ensures a finite spectral edge. However, they acknowledged that the analytical results are limited to uncorrelated Gaussian inputs.

- **Connection to training dynamics:** Reviewers Z5EM, upYX, and x8Zk questioned the link between the properties of the Hessian at initialisation and the actual success of GD training. *Rebuttal:* The authors added Appendix F, presenting preliminary results linking the BBP transition at initialisation to the onset of weak recovery in GD, confirming that overparameterization anticipates this transition.

- **Finite-size effects:** Reviewer W5zG highlighted mismatches between theory and simulations in the discontinuous BBP case and asked for better characterisation of finite-size corrections. *Rebuttal:* The authors explained that theoretically deriving finite-size corrections is prohibitive due to the resummation of non-perturbative diagrams.

- **Clarity:** Reviewer Z5EM and x8Zk asked for clarifications on terminology and methodology used in the paper. *Rebuttal:* The authors updated the draft and included a methodological overview in the main text.

In conclusion, most critiques were addressed. Remaining limitations concern generalisability to non-Gaussian inputs and and the limitation to preliminary results for training dynamics, though a comprehensive analysis can be considered beyond the scope.

**Reviewer Scores:**

- **Reviewer upYX Rating: 8 / Confidence: 3** The rebuttal addressed their main concern regarding the limitation to quadratic activations by adding an appendix on sigmoid/tanh activations. They also addressed (at least partially) the request for connections to optimisation dynamics. I would expect a confirmation of the rating.

- **Reviewer W5zG Rating: 8 / Confidence: 4** This reviewer explicitly replied to the rebuttal, stating they were *"fully satisfied"* with the answers, confirming rating and confidence.

- **Reviewer x8Zk Rating: 6 / Confidence: 4** Their main critique was the lack of a methodological overview in the main text and missing references. The authors updated the draft to include this overview in Section 3 and added the citations. I would expect an increase of the rating.

- **Reviewer Z5EM Rating: 4 / Confidence: 2** This reviewer had low confidence and the rebuttal clarified the terminology and added the requested experiments linking the Hessian to training dynamics (the primary weaknesses reported in the review). I would expect a rating increase, likely to 6.

---

### Decision · Program_Chairs · 2026-01-26

Accept (Oral)